# *CFTR* mRNAs with nonsense codons are degraded by the SMG6-mediated endonucleolytic decay pathway

Edward J. Sanderlin[1], Melissa M. Keenan[1], Martin Mense [2], Alexey S. Revenko [1], Brett P. Monia[1], Shuling Guo[1] & Lulu Huang [1✉]

Approximately 10% of cystic fibrosis patients harbor nonsense mutations in the *cystic fibrosis transmembrane conductance regulator* (*CFTR*) gene which can generate nonsense codons in the *CFTR* mRNA and subsequently activate the nonsense-mediated decay (NMD) pathway resulting in rapid mRNA degradation. However, it is not known which NMD branches govern the decay of *CFTR* mRNAs containing nonsense codons. Here we utilize antisense oligonucleotides targeting NMD factors to evaluate the regulation of nonsense codon-containing *CFTR* mRNAs by the NMD pathway. We observe that *CFTR* mRNAs with nonsense codons G542X, R1162X, and W1282X, but not Y122X, require UPF2 and UPF3 for NMD. Furthermore, we demonstrate that all evaluated *CFTR* mRNAs harboring nonsense codons are degraded by the SMG6-mediated endonucleolytic pathway rather than the SMG5-SMG7-mediated exonucleolytic pathway. Finally, we show that upregulation of all evaluated *CFTR* mRNAs with nonsense codons by NMD pathway inhibition improves outcomes of translational readthrough therapy.

[1] Ionis Pharmaceuticals, Inc., Carlsbad, CA, USA. [2] Cystic Fibrosis Foundation Therapeutics Lab, Cystic Fibrosis Foundation, Lexington, MA, USA.
✉email: lhuang@ionisph.com

Cystic fibrosis (CF) is a monogenetic autosomal recessive disorder caused by loss-of-function mutations in the *cystic fibrosis transmembrane conductance regulator* (*CFTR*) gene[1–3]. Approximately 10% of the cystic fibrosis population have at least one *CFTR* allele with a nonsense mutation that generates a nonsense codon (premature termination codon) in the mRNA[4–6]. Nonsense mutations are often associated with more severe CF phenotypes due to significantly reduced CFTR expression as a result of both RNA degradation by the nonsense-mediated decay (NMD) pathway and protein truncation[7,8]. Efforts in developing therapies for CF caused by nonsense mutations have focused on identifying small molecule readthrough drugs that suppress translation termination at the nonsense codon to produce full-length CFTR protein. Currently, no readthrough therapy has been approved for the treatment of CF caused by nonsense mutations. This non-success is in part because of the low abundance of mRNA substrates available for translational readthrough due to degradation by the NMD pathway[9,10]. Combination therapy will be required to inhibit NMD for the upregulation of mRNAs with nonsense codons while simultaneously promoting translational readthrough. Inhibition of NMD alone may also improve disease phenotypes provided the truncated proteins are functional[11].

The NMD pathway is not only an RNA surveillance pathway that degrades mRNAs with nonsense codons, but also a post-transcriptional regulatory mechanism governing ~10–20% of the normal transcriptome[12]. Therefore, the safety of pharmacological inhibition of the NMD pathway requires thorough investigation. Several lines of evidence suggest that NMD is not a single biochemical pathway in higher eukaryotes, but rather a pathway with several branches[12,13]. Three branches of the NMD pathway, diverging at the stage of nonsense codon recognition, have been reported including UPF2-independent, UPF3B-independent, and EJC-independent branches[12–17]. At the step of RNA destruction, several studies reveal that NMD substrate RNAs can be degraded through either SMG6-mediated endonucleolytic degradation or SMG5-SMG7-mediated exonucleolytic degradation[18–20]. These branch-specific NMD factors could potentially be safer therapeutic targets for diseases caused by nonsense mutations as each of them regulates a subset of endogenous NMD substrates[12,13,21]. For example, we have previously shown ASO-mediated depletion of branch-specific NMD factor UPF3B was well tolerated in mice with minimal impact on the normal transcriptome while alleviating disease phenotypes in a hemophilia mouse model[21].

However, it is not known which branches of the NMD pathway regulate *CFTR* mRNAs with nonsense codons and if the location of the nonsense codon influences branch-specificity. A better understanding of how the NMD pathway regulates *CFTR* mRNAs containing nonsense codons could provide a path for the development of NMD inhibition therapy for CF caused by nonsense mutations. Here, we sought to use antisense oligonucleotides (ASOs) to identify branch-specific NMD factors whose expression could be reduced to effectively impede NMD and upregulate the expression of *CFTR* mRNAs with nonsense codons to alleviate CF disease phenotypes.

Antisense technology is a clinically validated drug discovery platform[22]. ASOs are short synthetic oligonucleotides which bind specifically to their RNA targets through Watson-Crick base pairing to form ASO-RNA heteroduplexes. These DNA-RNA heteroduplexes can be substrates for the ubiquitous endonuclease RNase H, which can lead to the degradation of the target RNA[22,23]. We used ASOs to reduce the expression of NMD pathway components so as to identify branch-specific NMD factors which govern the NMD of *CFTR* mRNAs containing prevalent CFTR nonsense codons.

## Results

### CFF-16HBEge cell lines harboring *CFTR* nonsense alleles have reduced CFTR expression and function.

We employed the CFF-16HBEge cell model system to evaluate NMD regulation of *CFTR* mRNAs harboring various nonsense codons[24]. CFF-16HBEge cell lines were generated from 16HBE14o-cells by CRISPR-mediated genome editing and are an ideal cellular model system to study NMD as the *CFTR* nonsense mutations exist within their native genomic context and are under the regulation of the endogenous *CFTR* promoter. We grouped CFTR nonsense mutations into three categories as either early nonsense mutations—which generate nonsense codons within the first 150 amino acids, middle nonsense mutations—which generate nonsense codons between the 150th and 1000th amino acid, or late nonsense mutations—which generate nonsense codons after the 1000th amino acid. We selected CFTR nonsense mutations with the highest allele frequency from each category. As such, CFF-16HBEge cells containing CFTR nonsense mutations Y122X (early), G542X (middle), R1162X (late), and W1282X (late) were employed for evaluation of NMD pathway regulation of CFTR mRNAs (Fig. 1a).

We first evaluated CFTR expression and function in both 16HBE14o- (parental) and CFF-16HBEge cells. We found that levels of *CFTR* mRNAs from CFF-16HBEge-Y122X, -G542X, -R1162X, and -W1282X cell lines were reduced by 50-80% when compared to the levels of wild-type *CFTR* mRNAs in parental cells (Fig. 1b). The levels of *CFTR* mRNAs in CFF-16HBEge-G542X and -W1282X cells were consistent with previous reports[24–26]. We were unable to detect full-length CFTR proteins in any of these cell lines by western blot analysis (Fig. 1c). However, the presence of the partially stable, C-truncated W1282X-CFTR protein was detected (Supplementary Fig. 1), as previously characterized[25,27–30]. CFTR functional evaluation by Ussing chamber assays revealed minimal basal CFTR chloride channel function of <1 µA/cm² in CFF-16HBEge cells containing nonsense alleles, whereas that in the parental cells was 118 µA/cm² (Fig. 1d, e). CFF-16HBEge-Y122X cells showed higher CFTR mRNA levels of ~50% of parental cells and higher CFTR function of ~1 µA/cm² when compared to other CFF-16HBEge cell lines (Fig. 1b–e). Taken together, CFF-16HBEge cells with CFTR nonsense alleles demonstrated significantly reduced CFTR expression and function.

### ASOs mediate efficient NMD factor reduction and NMD pathway inhibition.

Efficacious ASOs were generated to reduce the expression of NMD factors UPF1, UPF2, UPF3A, UPF3B, SMG1, SMG5, SMG6, SMG7, SMG8, or SMG9 (Fig. 1f). Cells were treated with unformulated ASOs by free uptake as previously described[31]. RT-qPCR analysis demonstrated efficient reduction of NMD factor mRNAs following ASO treatment, reaching between 87% and 99% target reduction when compared to PBS and a negative control ASO treated samples (Fig. 2, Supplementary Figs. 2–3).

The mRNAs encoding NMD factors are themselves targets of NMD and exist under negative feedback regulation by the NMD pathway[12,32,33]. ASO-mediated reduction of NMD factors UPF1, UPF2, SMG1, and SMG6 resulted in robust upregulation of other NMD factor mRNAs (Fig. 2a, b, e, g), indicative of effective NMD pathway inhibition. ASO-mediated reduction of NMD factors UPF3B, SMG5, SMG7, SMG8, and SMG9 resulted in modest inhibition of the NMD pathway (Fig. 2d, f, h–j). Finally, reduction of NMD factor UPF3A did not upregulate any evaluated endogenous NMD substrate in a dose-responsive manner (Fig. 2c). Treatment of 16HBE cells with a negative control ASO did not significantly affect NMD factor mRNA levels (Supplementary

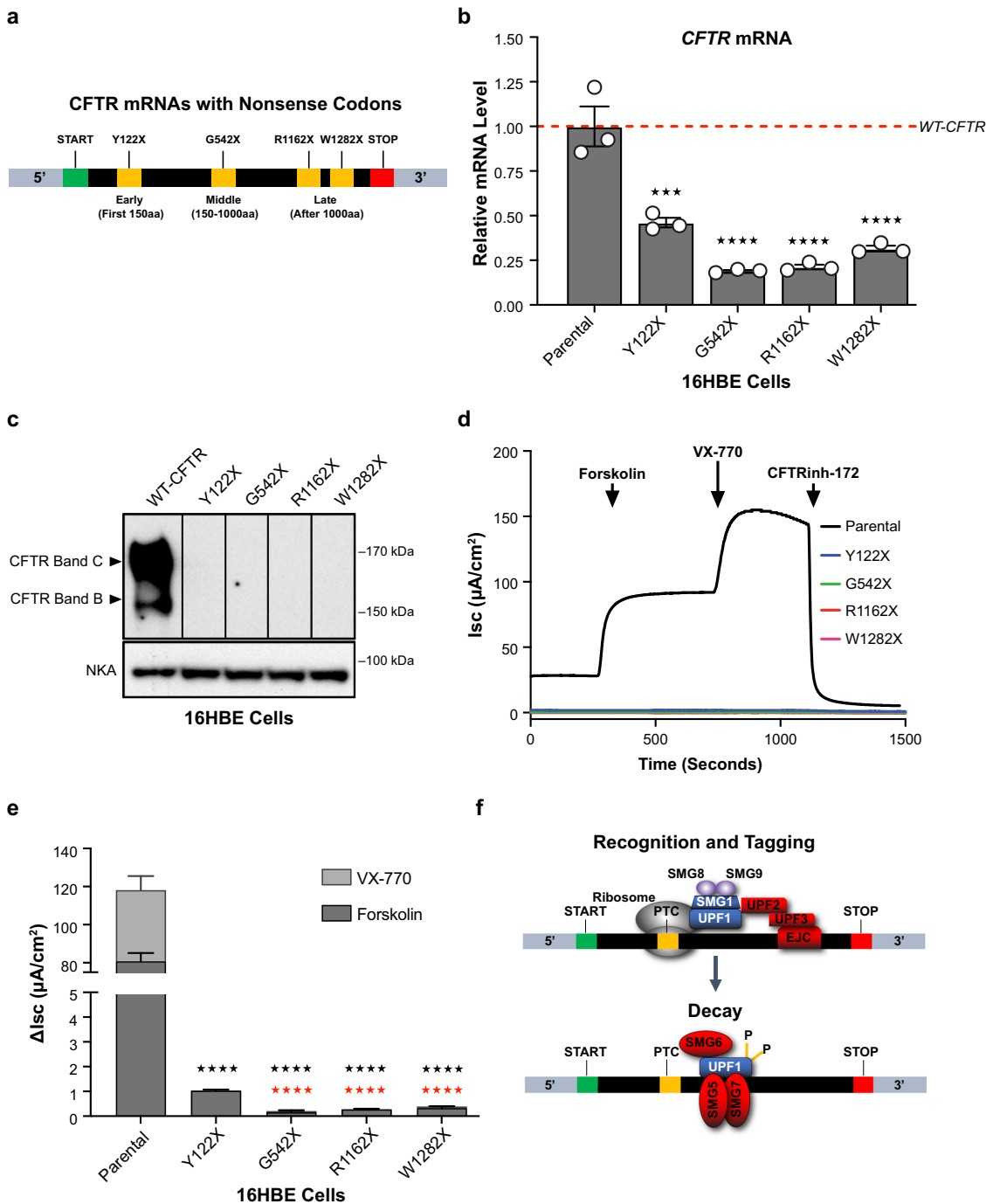

**Fig. 1 Characterization of the CFF-16HBEge in vitro model system.** CFF-16HBEge cells with *CFTR* Y122X, G542X, R1162X, and W1282X nonsense mutations have significantly reduced CFTR expression and function when compared to parental cells. **a** A schematic displaying *CFTR* nonsense mutations which were evaluated in this study. **b** RT-qPCR analysis of *CFTR* mRNAs in parental and CFF-16HBEge cells with Y122X, G542X, R1162X, or W1282X *CFTR* nonsense mutations. *CFTR* mRNA levels were normalized to total RNA levels ($n = 3$ for each group). The wild-type (WT) *CFTR* mRNA level in parental cells was set to 1 and is indicated by the dotted red line. **c** Representative western blot image of CFTR protein expression in parental and CFF-16HBEge cells. Na$^+$/K$^+$-ATPase (NKA) protein was used as a loading control ($n = 3$ for each group). **d** Representative short circuit current (Isc) traces of parental and CFF-16HBEge cells obtained using the Ussing Chamber assay. 7 μM benzamil was added to establish baseline current followed by the sequential addition of 20 μM forskolin, 1 μM VX-770, and 20 μM CFTR inhibitor-172 (CFTRinh-172) at times indicated by arrows. **e** ΔIsc of CFTR-attributed current in parental and CFF-16HBEge cells ($n = 4$ for parental, $n = 5$ for CFF-16HBEge-Y122X, $n = 3$ for CFF-16HBEge-G542X, $n = 4$ for CFF-16HBEge-R1162X, and $n = 5$ for CFF-16HBEge-W1282X cells). **f** A schematic displaying the recognition, tagging, and decay steps of the NMD pathway. Core NMD factors are indicated by the color blue, SMG1-cofactors SMG8 and SMG9 are indicated by the color purple, and branch-specific NMD components are colored red. Data are presented as mean ± SEM from biologically independent samples. Statistical significance was analyzed by one-way ANOVA followed by Dunnett's multiple-comparison test (***$p < 0.001$, ****$p < 0.0001$). Black stars indicate statistical comparisons made between parental cells and CFF-16HBEge cells. Red stars indicate statistical comparisons made between CFF-16HBEge-Y122X and CFF-16HBEge-G542X, -R1162X, and -W1282X cells. See Source Data file for the exact *p*-values. Source data are provided as a Source Data file.

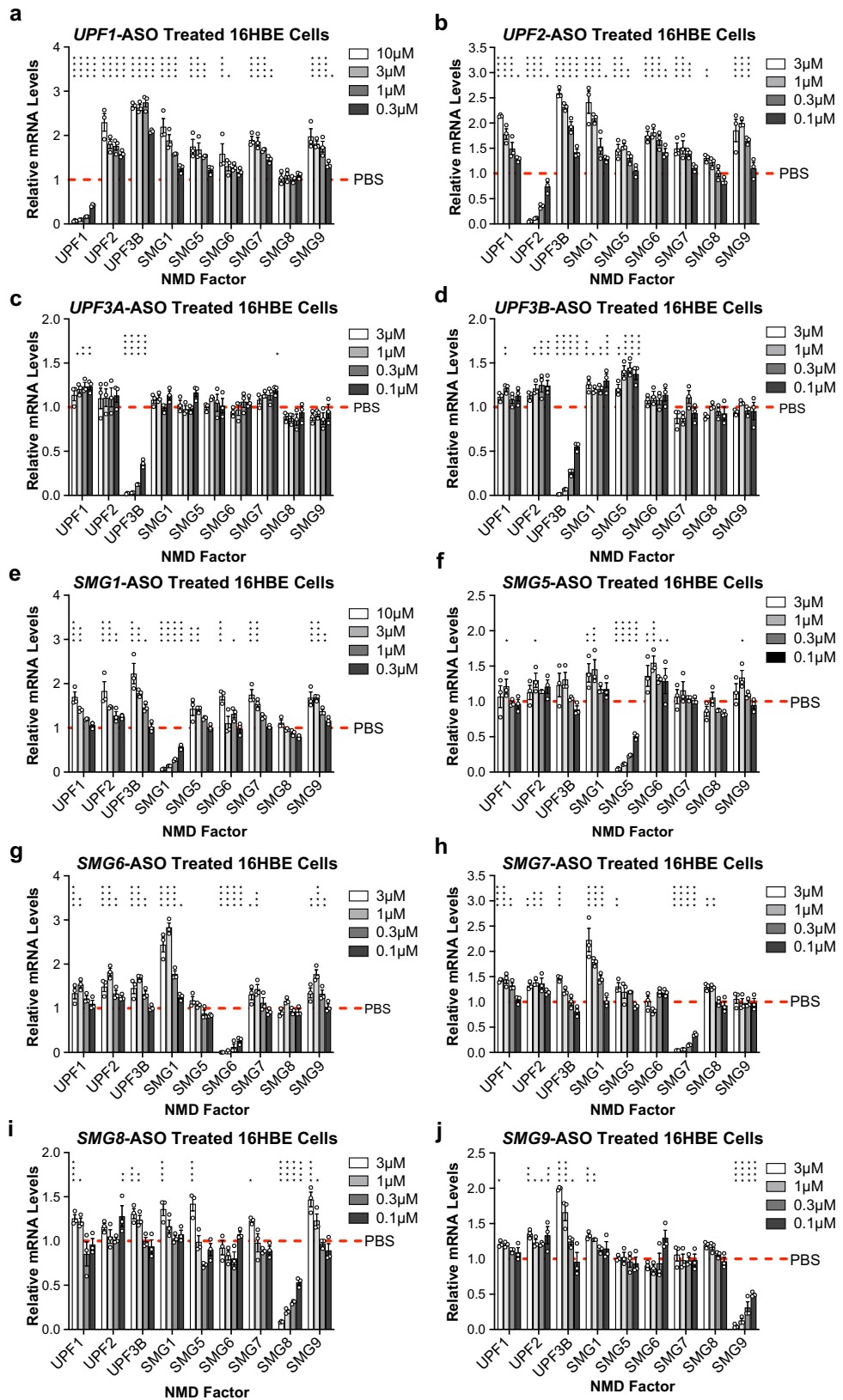

Fig. 3). Furthermore, ASO-mediated reduction of NMD factors had no significant effects on wild-type *CFTR* mRNA levels in parental cells (Supplementary Fig. 4). ASO treatments in CFF-16HBEge cells with Y122X, G542X, R1162X, and W1282X nonsense mutations reflected similar levels of NMD factor target reduction and corresponding NMD pathway inhibition as was observed in parental cells for each ensuing experiment.

**All evaluated *CFTR* mRNAs with nonsense codons are regulated by UPF1 and SMG1.** To validate *CFTR* mRNAs in CFF-

**Fig. 2 Treatment of parental 16HBE cells with ASOs targeting NMD pathway components resulted in efficient target reduction and NMD pathway inhibition.** RT-qPCR analysis of NMD factor mRNA levels in parental cells treated with ASOs targeting (**a**) *UPF1*, (**b**) *UPF2*, (**c**) *UPF3A*, (**d**) *UPF3B*, (**e**) *SMG1*, (**f**) *SMG5*, (**g**) *SMG6*, (**h**) *SMG7*, (**i**) *SMG8*, or (**j**) *SMG9*. NMD factor mRNA levels were normalized to total RNA levels. NMD factor mRNA levels were also normalized to housekeeping gene *β-actin* mRNA (Supplementary Fig. 2-3). PBS control levels were set to 1 and are indicated by dotted red lines. $n = 3$ for each group. Data are presented as mean ± SEM from biologically independent samples. Statistical significance was analyzed by two-way ANOVA followed by Dunnett's multiple-comparison test (\*$p < 0.05$, \*\*$p < 0.01$, \*\*\*$p < 0.001$, \*\*\*\*$p < 0.0001$). See Source Data file for the exact *p*-values. Source data are provided as a Source Data file.

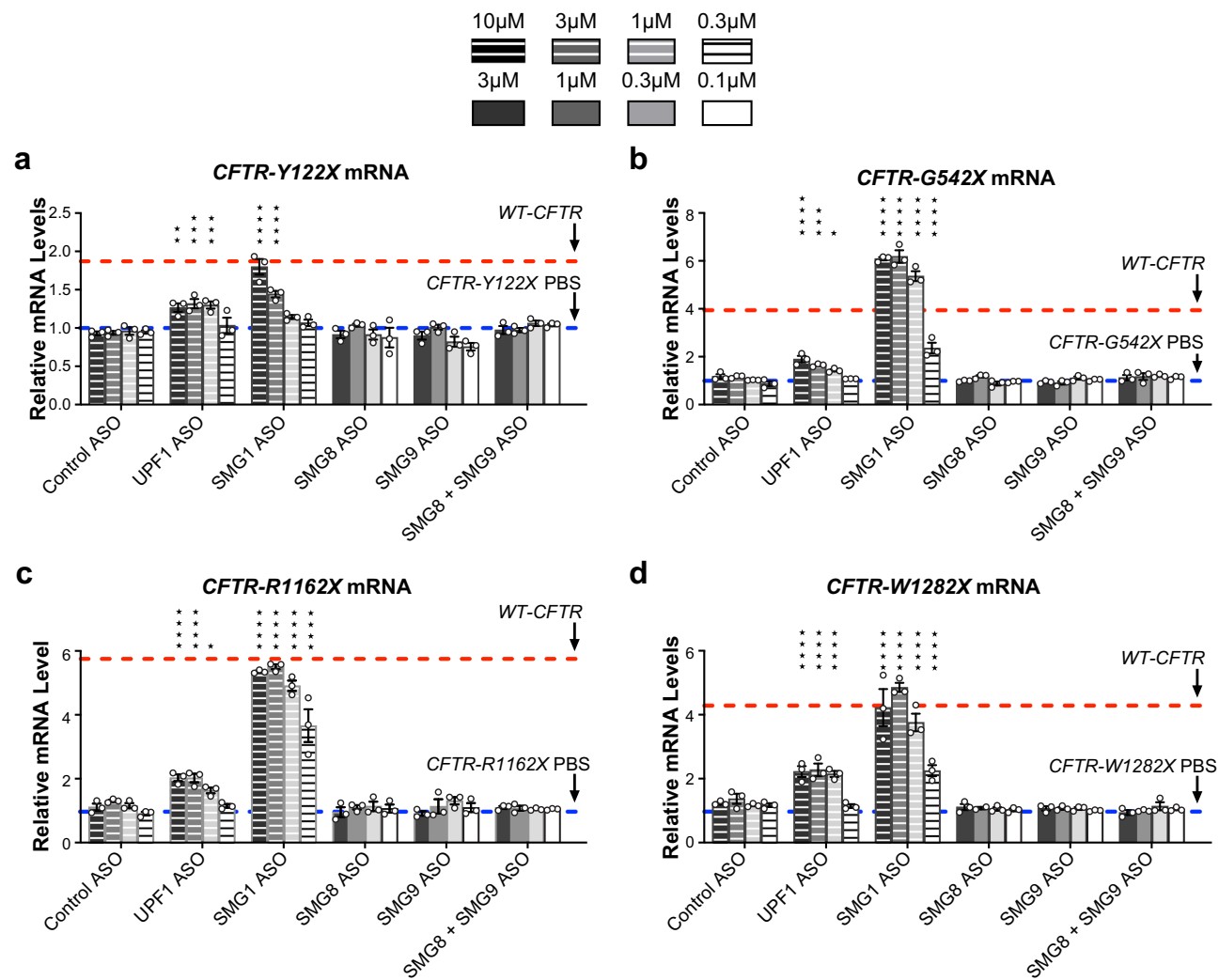

**Fig. 3 *CFTR* mRNAs containing nonsense codons are regulated by UPF1 and SMG1 but not SMG8 or SMG9.** RT-qPCR analysis of *CFTR* mRNAs in CFF-16HBEge cells with (**a**) Y122X, (**b**) G542X, (**c**) R1162X, or (**d**) W1282X *CFTR* nonsense mutations treated with ASOs targeting *UPF1*, *SMG1*, *SMG8*, or *SMG9*. Cells treated with PBS or a scrambled control ASO were used as negative controls. *CFTR* mRNA levels were normalized to total RNA levels. *CFTR* mRNA levels were also normalized to housekeeping gene *β-actin* mRNA (Supplementary Fig. 5). PBS control levels were set to 1 and are indicated by dotted blue lines. Wild-type (WT) *CFTR* mRNA levels in parental cells are indicated by dotted red lines. $n = 3$ in each group. Data are presented as mean ± SEM from biologically independent samples. Statistical significance was analyzed by two-way ANOVA followed by Dunnett's multiple-comparison test (\*$p < 0.05$, \*\*$p < 0.01$, \*\*\*$p < 0.001$, \*\*\*\*$p < 0.0001$). See Source Data file for the exact *p*-values. Source data are provided as a Source Data file.

16HBEge cells are NMD targets, we treated these cells with ASOs targeting core NMD factors *UPF1* and *SMG1*. We found that *UPF1*- or *SMG1*-ASO treatment led to upregulation of all evaluated *CFTR* mRNAs with nonsense codons, which indicate these transcripts are bona fide NMD targets (Fig. 3, Supplementary Fig. 5). *UPF1* depletion resulted in 1.5- to 2.5-fold upregulation of *CFTR* mRNAs with nonsense codons (Fig. 3, Supplementary Fig. 5). Interestingly, *SMG1* depletion resulted in a more robust upregulation of all *CFTR* mRNAs with nonsense codons which reached levels comparable to wild-type *CFTR* mRNA levels in

parental cells (Fig. 3, Supplementary Fig. 5). However, ASOs targeting SMG1 regulatory cofactors *SMG8* or *SMG9*, or both in combination, did not upregulate any evaluated *CFTR* mRNAs with nonsense codons (Fig. 3, Supplementary Fig. 5).

***CFTR* mRNAs with nonsense codons G542X, R1162X, and W1282X, but not Y122X, are recognized by UPF2 and UPF3.** mRNAs with nonsense codons can be recognized by different branches of the NMD pathway[12,14–17]. We used ASO-mediated depletion of NMD branch-specific factors to examine which

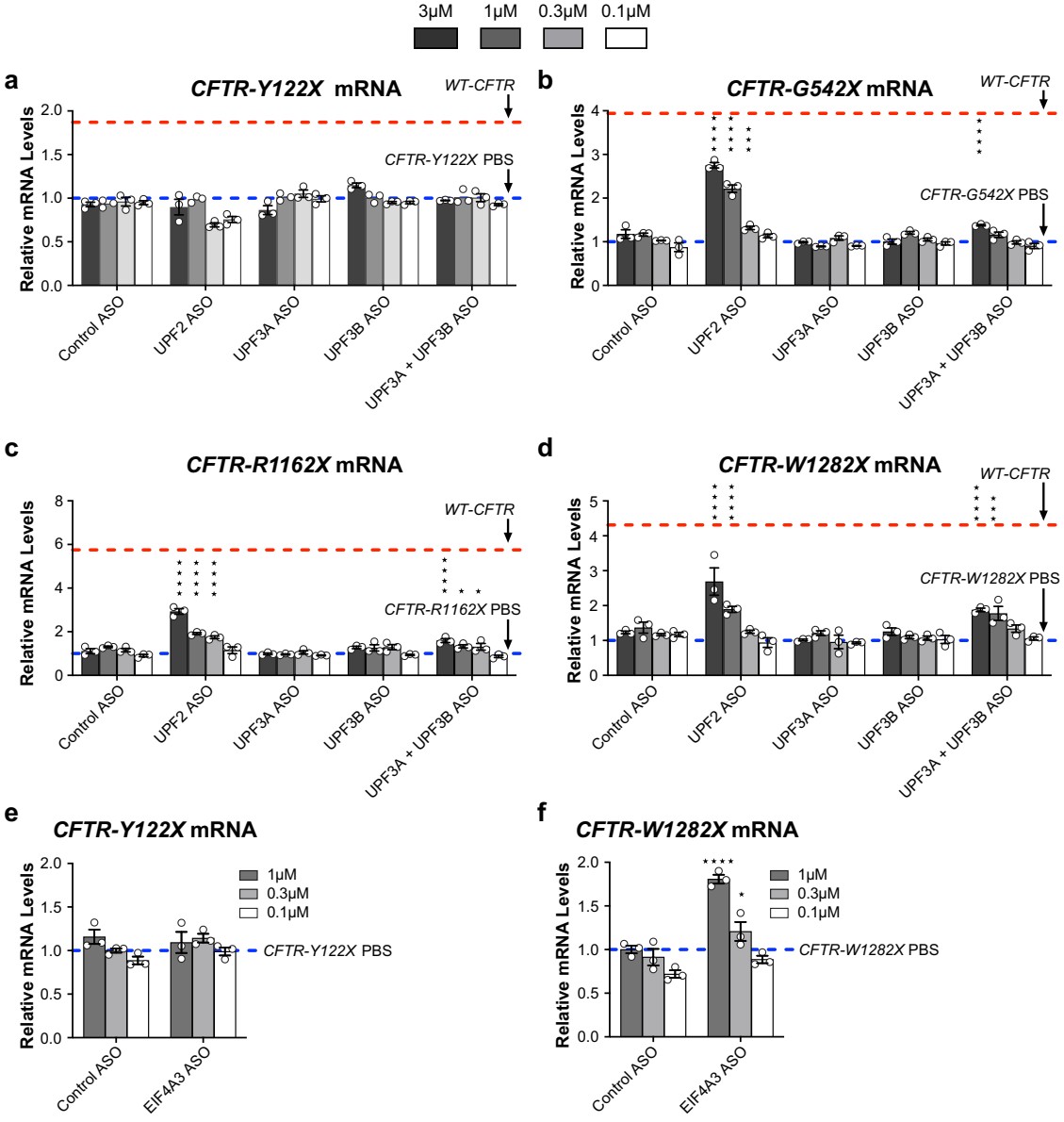

**Fig. 4 *CFTR* mRNAs with nonsense codons G542X, R1162X, and W1282X, but not Y122X, are recognized by UPF2 and UPF3 paralogs.** RT-qPCR analysis of CFF-16HBEge cells with (**a**) Y122X, (**b**) G542X, (**c**), R1162X, or (**d**) W1282X *CFTR* nonsense mutations treated with ASOs targeting *UPF2*, *UPF3A*, or *UPF3B*. RT-qPCR analysis of CFF-16HBEge cells with (**e**) Y122X or (**f**) W1282X *CFTR* nonsense mutations treated with ASO targeting *eIF4A3*. Cells treated with PBS or a scrambled control ASO were used as negative controls. *CFTR* mRNA levels were normalized to total RNA levels. *CFTR* mRNA levels were also normalized to housekeeping gene *β-actin* mRNA (Supplementary Fig. 5). PBS control levels were set to 1 and are indicated by blue dotted lines. Wild-type (WT) *CFTR* mRNA levels from parental cells are indicated by red dotted lines. $n = 3$ for each group. Data are presented as mean ± SEM from biologically independent samples. Statistical significance was analyzed by two-way ANOVA followed by Dunnett's multiple-comparison test ($*p < 0.05$, $***p < 0.001$, $****p < 0.0001$). See Source Data file for the exact *p*-values. Source data are provided as a Source Data file.

NMD branch recognizes *CFTR* mRNAs with various nonsense codons. CFF-16HBEge cells were treated with ASOs targeting *UPF2*, *UPF3A*, or *UPF3B,* followed by analysis of NMD factor target reduction and *CFTR* mRNA upregulation by RT-qPCR. We observed that *UPF2*-ASO treatment upregulated *CFTR-G542X, -R1162X,* and *-W1282X* mRNAs to the levels that were ~50% of wild-type *CFTR* mRNA levels observed in parental cells; however, *CFTR-Y122X* mRNA was not upregulated by *UPF2*-ASO treatment (Fig. 4a–d, Supplementary Fig. 5). Furthermore, we observed depletion of either *UPF3A* or *UPF3B* did not significantly upregulate any *CFTR* mRNAs containing nonsense codons in CFF-16HBEge cells. However, combination treatment of *UPF3A*- and *UPF3B*-ASOs upregulated *CFTR-G542X,*

*-R1162X,* and *-W1282X* mRNA, but not *CFTR-Y122X* mRNA, in a dose-responsive manner when compared to the PBS control (Fig. 4a–d, Supplementary Fig. 5). These results suggest that *CFTR* mRNAs with nonsense codons G542X, R1162X, and W1282X, but not Y122X, are recognized by NMD factors UPF2 and UPF3, and that UPF3A and UPF3B paralogs are functionally redundant in regulating *CFTR* mRNAs.

One function for UPF2 and UPF3 proteins is to bridge UPF1 to the EJC to stimulate UPF1 helicase activity[34]. Therefore, we evaluated if the EJC is required for NMD of *CFTR* mRNAs harboring nonsense codons by using ASO that depletes the EJC core component *eIF4A3* (Supplementary Fig. 6). In CFF-16HBEge cells treated with *eIF4A3*-ASO, *CFTR* mRNAs with

nonsense codon W1282X, but not Y122X, were upregulated (Fig. 4e, f). Taken together, these results suggest that unlike other CFTR mRNAs with nonsense codons, the degradation of CFTR-Y122X mRNA is not regulated by UPF2, UPF3 or EJC proteins.

**All evaluated *CFTR* mRNAs with nonsense codons are degraded by the SMG6-mediated endonucleolytic decay pathway, but not the SMG5-SMG7-mediated exonucleolytic decay pathway**. After nonsense codon recognition, destruction of CFTR mRNAs could be arbitrated by either the SMG6-mediated endonucleolytic, SMG5-SMG7-mediated exonucleolytic, or a combination of both decay pathways[18,19,35]. We found that the SMG6-ASO significantly upregulated all evaluated CFTR mRNAs harboring nonsense codons to levels comparable to wild-type CFTR mRNA levels in parental cells (Fig. 5a–d, Supplementary Fig. 5). Maximal upregulation of CFTR mRNAs with nonsense codons was achieved when ASO-mediated SMG6 mRNA reduction was ~90% (Fig. 5a–d, Supplementary Fig. 5). In contrast, SMG5- or SMG7-ASO alone or in combination did not upregulate any evaluated CFTR mRNAs with nonsense codons despite achieving >95% target reduction (Fig. 5a–d, Supplementary Fig. 5).

We next confirmed SMG6-dependent degradation of CFTR mRNAs using primary human bronchial epithelial cells homozygous for the W1282X nonsense mutation (hBE W1282X/W1282X). Efficient ASO-mediated depletion of SMG6 mRNA by ~90% in hBE W1282X/W1282X cells resulted in a significant upregulation of CFTR mRNA by 3.4-fold when compared to PBS control (Fig. 5e, Supplementary Fig. 7). Treatment of SMG5- or SMG7-ASO alone or in combination did not significantly upregulate CFTR mRNAs in hBE W1282X/W1282X cells (Fig. 5e, Supplemental Fig. 7). Collectively, these data suggest that CFTR mRNAs harboring nonsense codons are predominately degraded via the SMG6-mediated endonucleolytic pathway rather than the SMG5-SMG7-mediated exonucleolytic pathway.

**NMD inhibition improves translational readthrough therapy**. Translational readthrough of nonsense codons, although it occurs at low frequency, can result in full-length protein that can improve disease phenotypes[36–40]. Therefore, we sought to evaluate if upregulation of CFTR mRNAs with nonsense codons induced by depletion of NMD factors can result in meaningful increases of full-length CFTR protein expression and function for phenotype rescue in CFF-16HBEge cells.

CFF-16HBEge cells were cultured at liquid–liquid interface and were treated with either SMG1- or SMG6-ASOs for 6 days. Although efficient depletion of SMG1 or SMG6 was achieved at both the mRNA and protein levels, the upregulation of CFTR mRNAs did not lead to detectable increases of full-length CFTR protein in all tested CFF-16HBEge cells harboring CFTR nonsense mutations (Fig. 6; Supplementary Figs. 8–11). We also could not detect increases of truncated CFTR proteins or CFTR functional improvement in CFF-16HBEge-Y122X, -G542X, or -R1162X cells when compared to PBS control (Figs. 6a–f, 7a–f). Interestingly, in CFF-16HBEge-W1282X cells both SMG1- and SMG6-ASO treatments significantly increased the level of C-truncated CFTR protein by nearly 35-fold when compared to PBS control (Fig. 6g, h), which resulted in increases of CFTR function between 2.6-fold and 4.2-fold, respectively (Fig. 7g, h).

We then tested if the upregulation of CFTR nonsense codon-containing mRNAs, which were induced by NMD inhibition, could improve the outcome of translational readthrough therapy. Cells were treated with SMG1- or SMG6-ASO for 6 days to inhibit NMD followed by the addition of aminoglycoside geneticin (G418; 100 μM) during the final 2 days of ASO treatments. We found that the combinations of SMG1- or SMG6-ASO with G418 increased full-length CFTR protein production in CFF-16HBEge-Y122X, -G542X, and -R1162X cells (Fig. 6a–f). Consistent with the observed CFTR protein upregulation, combination treatment with SMG1- or SMG6-ASO improved CFTR function by 1.6 to 2-fold when compared to G418 treatment alone in CFF-16HBEge cells containing Y122X, G542X, and R1162X nonsense mutations (Figs. 6a–f, 7a–f). Interestingly, unlike other CFTR nonsense mutations, full-length W1282X-CFTR protein production could not be detected by western blot following ASO treatments with or without G418 (Fig. 6g, h), which is consistent with previous reports[24,27,28]. However, the combination of SMG1- or SMG6-ASO with G418 resulted in upregulation of truncated CFTR protein and subsequent improvement of W1282X-CFTR function by nearly 2.7-fold and 3.4-fold, respectively, when compared to G418 treatment alone (Figs. 6g, h, 7g, h). Collectively, these data suggest NMD inhibition can significantly improve the outcome of translational readthrough therapy and lead to CFTR protein and function upregulation.

## Discussion

In this study, we used ASOs to reduce the expression of each NMD pathway component to determine which NMD branches regulate the degradation of CFTR mRNAs containing nonsense codons. We confirmed that the core NMD factors UPF1 and SMG1 regulate all evaluated CFTR mRNAs with nonsense codons (Fig. 3). Interestingly, SMG1-ASO triggered a more robust upregulation of CFTR mRNAs than UPF1-ASO, suggesting SMG1 could be the rate-limiting factor for the degradation of CFTR nonsense codon-containing mRNAs in CFF-16HBEge cells. However, ASO-mediated reduction of SMG1 regulating factors SMG8 and SMG9 alone or in combination did not upregulate CFTR mRNAs in CFF-16HBEge cells. We and others have reported that SMG8 or SMG9 depletion have limited impacts on NMD of β-GLOBIN mRNA when compared to SMG1 depletion in either MHT or Hela cells[21,41]. However, another study found SMG8 depletion was sufficient for upregulating nonsense codon-containing COL6A2 mRNA in Ullrich disease fibroblasts[11]. Regulation of NMD substrates by NMD can be subject to cell type- and tissue-specific contexts, which could explain why SMG8 and SMG9 do not regulate degradation of CFTR mRNAs in CFF-16HBEge cells but are essential for the destruction of other transcripts harboring nonsense codons in other cell types[11,12].

We next evaluated if the recognition of CFTR nonsense codon-containing mRNAs requires UPF2, UPF3, or EJC factors by ASO-mediated reduction of these branch-specific NMD factors. We found that recognition of CFTR-W1282X mRNAs were regulated by UPF2, UPF3 paralogs, and the EJC factor eIF4A3. Conversely, levels of CFTR mRNAs with the Y122X nonsense codon were not affected by ASOs targeting UPF2, UPF3 paralogs, or eIF4A3; suggesting that these proteins are not required for the recognition of CFTR-Y122X mRNA for NMD.

A recent report using a HEK293 CFTR expression minigene (EMG) system showed CFTR mRNAs with Y122X can evade NMD, as CFTR-Y122X mRNA levels were similar to wild-type CFTR mRNA. The EMG system contains the entire coding region of CFTR and introns flanking exon 4, which possesses the Y122X nonsense codon[42]. Here, we provide evidence CFTR-Y122X mRNA is a bona fide NMD substrate, as depletion of NMD factors UPF1, SMG1, and SMG6 results in mRNA upregulation in CFF-16HBEge-Y122X cells. However, the level of Y122X mRNA is higher when compared to other CFTR nonsense codon containing mRNAs in CFF-16HBEge cell lines (Fig. 1), suggesting Y122X mRNA may partially evade NMD. It has been reported that translation re-initiation events occurring downstream of a

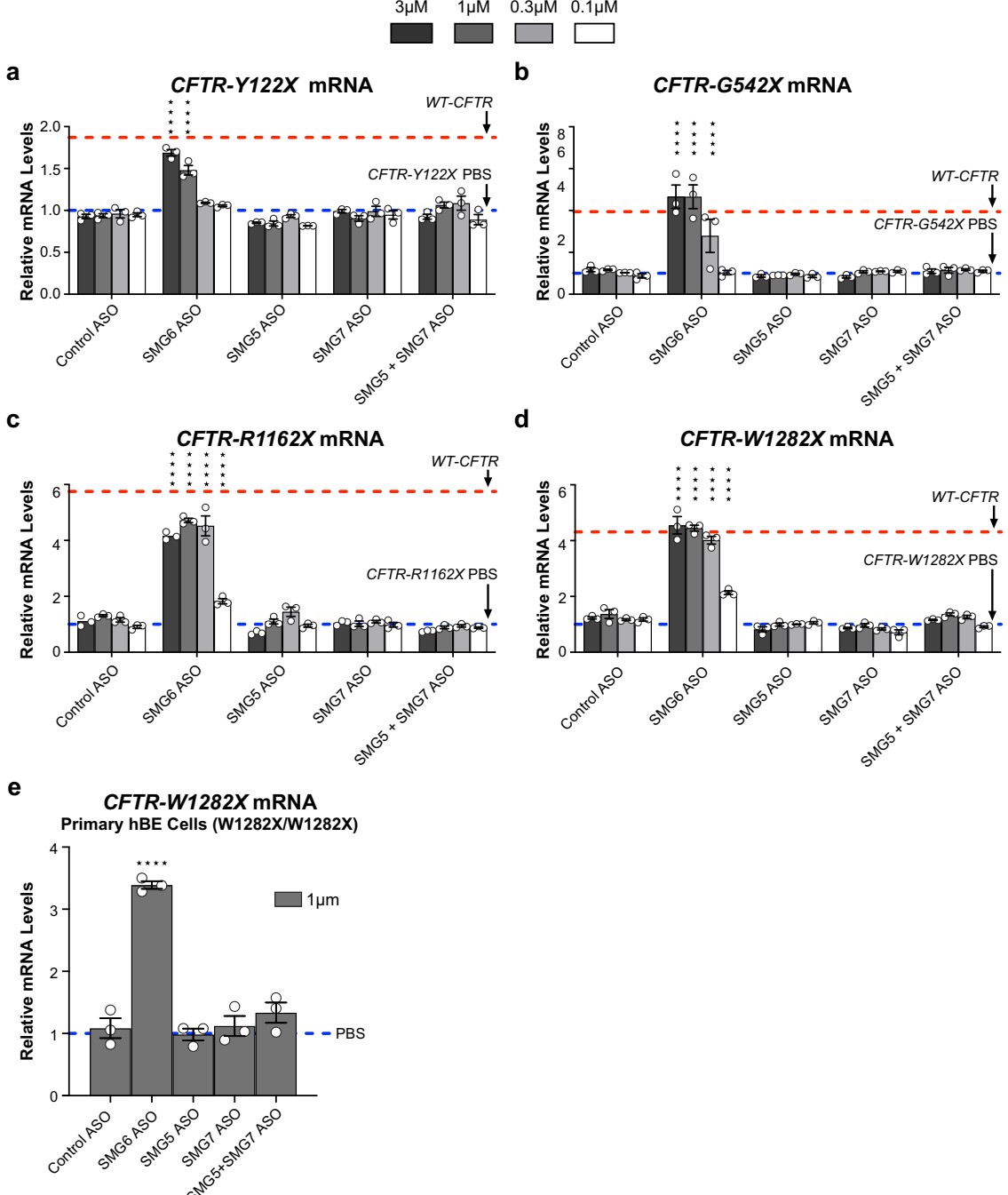

**Fig. 5 *CFTR* mRNA with nonsense codons are degraded by the SMG6-mediated endonucleolytic decay pathway but not the SMG5-SMG7-mediated exonucleolytic decay pathway.** RT-qPCR analysis of CFF-16HBEge cells with (**a**) Y122X, (**b**) G542X, (**c**) R1162X, or (**d**) W1282X *CFTR* nonsense mutations treated with ASOs targeting *SMG6*, *SMG5*, or *SMG7*. **e** RT-qPCR analysis of *CFTR* mRNA levels in hBE cells homozygous for the W1282X nonsense mutation treated with ASOs targeting *SMG6*, *SMG5*, or *SMG7*. Cells treated with PBS or a scrambled control ASO were used as negative controls. *CFTR* mRNA levels were normalized to total RNA levels in CFF-16HBEge and hBE cells. *CFTR* mRNA levels were also normalized to housekeeping gene *β-actin* mRNA (Supplementary Fig. 7b). PBS control levels for *CFTR* mRNA expression were set to 1 and are indicated by blue dotted lines. Wild-type (WT) *CFTR* mRNA levels from parental cells are indicated by red dotted lines. $n = 3$ for each group. Data are presented as mean ± SEM from biologically independent samples. Statistical significance was analyzed by (**a–d**) two-way ANOVA or (**e**) one-way ANOVA followed by Dunnett's multiple-comparison test (****$p < 0.0001$). See Source Data file for the exact $p$-values. Source data are provided as a Source Data file.

nonsense codon could protect mRNAs from NMD, presumably due to the removal of the EJC-UPF2-UPF3 complex[43]. Previous studies have found that methionine codons M150, M152, M156, and M265 in *CFTR* mRNA can mediate alternative translation initiation events resulting in the production of N-truncated CFTR protein products[44,45]. It is possible that translation re-initiation

events downstream of the Y122X nonsense codon partially protect *CFTR-Y122X* mRNA from NMD. In support of this hypothesis, we observed that the depletion of UPF2, UPF3 paralogs, or EJC factor eIF4A3 did not upregulate *CFTR-Y122X* mRNA in CFF-16HBEge-Y122X cells. Collectively, our results suggest that *CFTR-Y122X* mRNA is degraded by the NMD

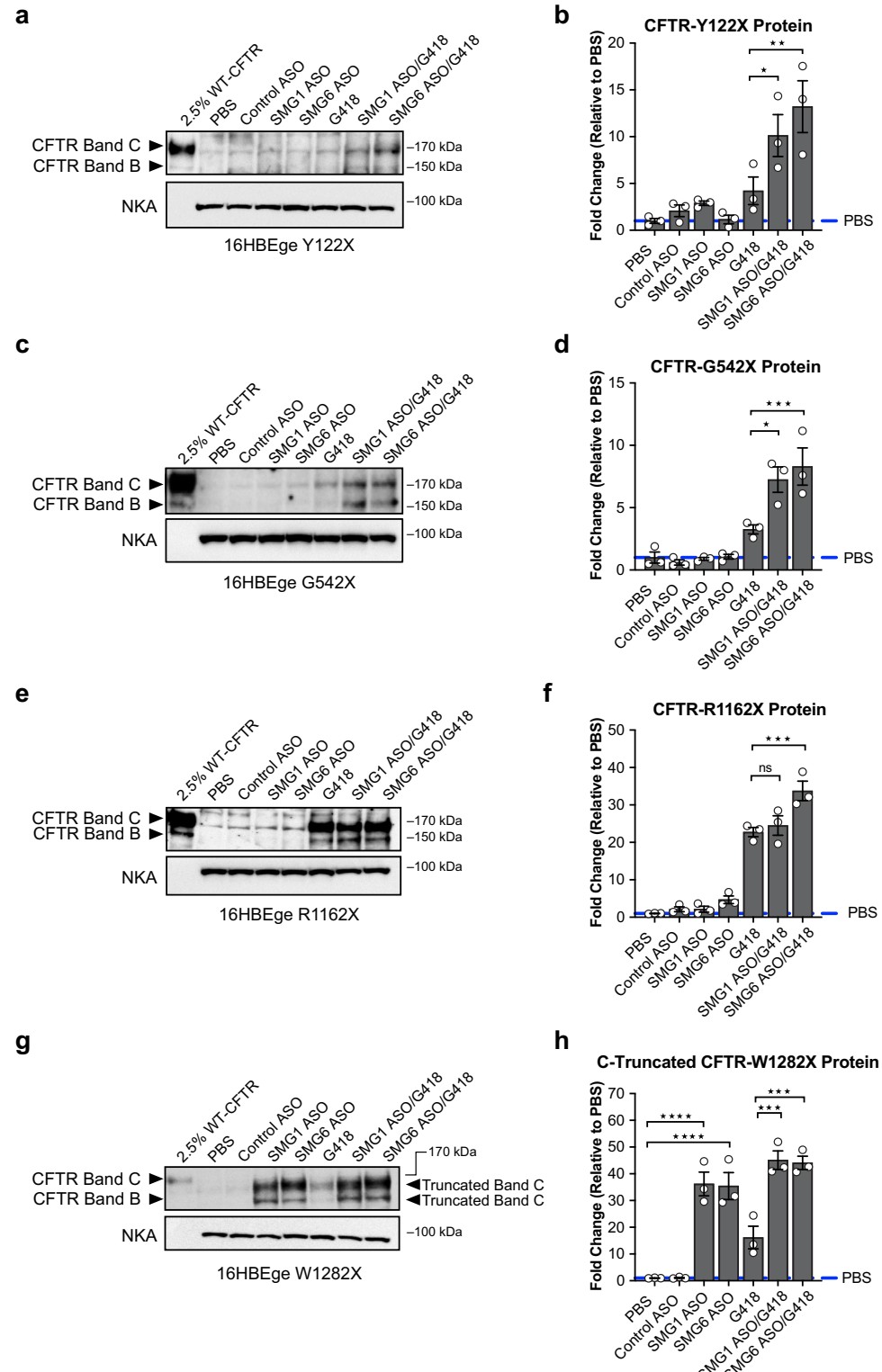

**Fig. 6 *SMG6*-ASO treatment improves translational readthrough outcomes and enhances CFTR protein upregulation in CFF-16HBEge cells.** Cells were treated with *SMG1*- or *SMG6*-ASO for 6 days followed by the addition of aminoglycoside geneticin (G418; 100 µM) during the final 2 days of ASO treatments. Representative western blot images (left) and quantification (right) of CFTR protein expression in CFF-16HBEge cells with (**a**, **b**) Y122X, (**c**, **d**) G542X, (**e**, **f**) R1162X, and (**g**, **h**) W1282X nonsense mutations following indicated treatment conditions. Full-length, complex-glycosylated band C of CFTR protein was evaluated for western blot quantification for (**b**) CFF-16HBEge-Y122X, (**d**) -G542X, (**f**) -R1162X cells. **h** C-truncated band C of CFTR-W1282X protein was quantified for CFF-16HBEge-W1282X cells. Na⁺/K⁺-ATPase (NKA) protein was used as a loading control. $n = 3$ for each group. Data are presented as mean ± SEM from biologically independent samples. Statistical significance was analyzed by one-way ANOVA followed by Sidak's multiple-comparison test (*$p < 0.05$, **$p < 0.01$, ***$p < 0.001$, ****$p < 0.0001$). See Source Data file for the exact $p$-values. Source data are provided as a Source Data file.

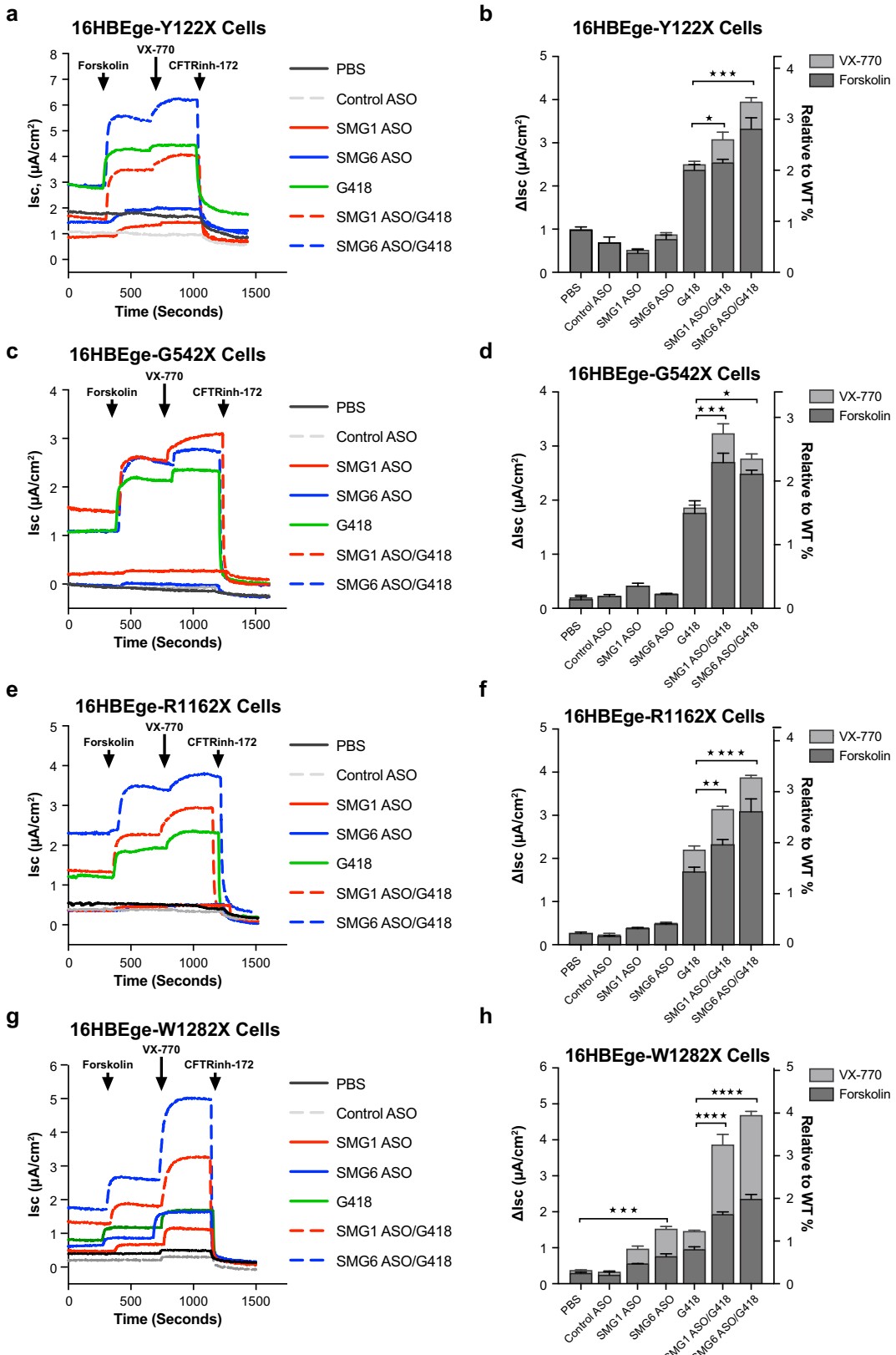

pathway independent of UPF2, UPF3 paralogs, and EJC factor eIF4A3. One advantage of the CFF-16HBEge cell model system used in this study is that the *CFTR Y122X* nonsense mutation exists within the native genomic context, which could more closely reflect the endogenous situation. However, further

investigation using primary patient cells containing the *CFTR Y122X* nonsense mutation is required.

After nonsense codon recognition and tagging by SMG1-mediated phosphorylation of UPF1, destruction of NMD substrates can occur through endonucleolytic cleavage by SMG6 or

**Fig. 7 SMG6-ASO treatment improves translational readthrough outcomes and enhances CFTR function improvement in CFF-16HBEge cells.** Cells were treated with SMG1- or SMG6-ASO for 6 days followed by the addition of aminoglycoside geneticin (G418; 100 μM) during the final 2 days of ASO treatments. Representative Isc traces (left) and ΔIsc (right) were obtained from Ussing chamber assays of CFF-16HBEge cells with (**a, b**) Y122X, (**c, d**) G542X, (**e, f**) R1162X, or (**g, h**) W1282X CFTR nonsense mutations following the indicated treatment conditions. For the Ussing chamber assay, 7 μM benzamil was added to establish baseline currents followed by the sequential addition of 20 μM forskolin, 1 μM VX-770, and 20 μM CFTR inhibitor-172 (CFTRinh-172) at times indicated by arrows to determine CFTR function. $n = 3$ for each group. Data are presented as mean ± SEM from biologically independent samples. Statistical significance was analyzed by one-way ANOVA followed by Sidak's multiple-comparison test (*$p < 0.05$, **$p < 0.01$, ***$p < 0.001$, ****$p < 0.0001$). See Source Data file for the exact $p$-values. Source data are provided as a Source Data file.

deadenylation and decapping elicited by the SMG5-SMG7 heterodimer[35,46,47]. Several lines of evidence suggest significant redundancy exists between the endo- and exonucleolytic decay routes as both nonsense codon-containing mRNAs and endogenous NMD substrates were shown to utilize both decay routes simultaneously[18,19,35]. Interestingly, we found ASO-mediated depletion of SMG6, but not SMG5 or SMG7, significantly upregulated all evaluated CFTR mRNAs regardless of the nonsense codon locations in the mRNA. A recent study provided evidence that some nonsense codon-containing NMD substrates can be preferentially routed through SMG6-dependent endonucleolytic decay rather than the SMG5-SMG7-dependent exonucleolytic decay pathway[19]. This report supports our observations as turnover of CFTR mRNAs containing nonsense codons are predominately arbitrated through the SMG6-mediated endonucleolytic route, but not the SMG5-SMG7-dependent exonucleolytic decay route. ASO-mediated depletion of SMG6 upregulated CFTR mRNAs with nonsense codons to a similar degree as the depletion of core NMD factors UPF1 and SMG1, suggesting SMG6-ASO could serve as a pan NMD inhibitor for CFTR mRNAs with various nonsense codons.

We next confirmed that CFTR W1282X mRNAs were highly responsive to NMD inhibition as we have previously reported[25], raising the possibility that NMD inhibition alone may be sufficient to improve cystic fibrosis disease phenotypes caused by the W1282X nonsense mutation. Interestingly, when combining both NMD inhibition and translational readthrough approaches, we observed augmented W1282X CFTR functional parameters when compared to corresponding levels of protein upregulation (Figs. 6 and 7). The functional improvement detected by the Ussing chamber assay could be attributed to the combination of both the upregulated C-terminal truncated protein and diminutive amounts of full-length protein induced by G418. Even though we are unable to detect full-length CFTR protein expression following G418 treatment by western blot, minute amounts of full-length CFTR protein could contribute a significant portion of the observed functional improvement.

However, we found that despite achieving robust upregulation of CFTR mRNAs in CFF-16HBEge cells following ASO-mediated SMG6 depletion, the magnitudes of corresponding CFTR protein and functional improvement differed between CFF-16HBEge cells following G418 treatment. Previous reports have suggested that the identity and sequence context of the PTC can influence translational readthrough efficiency of aminoglycosides and thereby affect the levels of the full-length readthrough product[37]. Moreover, these same elements also affect the incorporation of different near-cognate tRNAs, which could impact maturation, stability, and function of the readthrough product[48,49]. These observations highlight the importance and challenge of identifying tolerable and efficacious translational readthrough compounds for combination with NMD inhibition therapy[8,24,37]. An investigational aminoglycoside derivative ELX-02 has shown promise in clinical trials and was given fast track designation by the U.S. Food and Drug Administration for people with cystic fibrosis caused by nonsense mutations[50,51]. We and others have

demonstrated that targeting translation termination factors in combination with aminoglycosides could be a promising therapeutic strategy for diseases caused by nonsense mutations[52,53].

Overall, our results advance the understanding of how the NMD pathway regulates CFTR mRNAs containing nonsense codons and provides the direction for the development of therapeutic NMD inhibitors for cystic fibrosis caused by nonsense mutations. Our findings suggest inhibition of the branch-specific NMD factor SMG6 could be a potential therapeutic approach for the upregulation of CFTR mRNAs harboring prevalent nonsense codons. We have previously reported that ASO-mediated reduction of branch-specific NMD factor UPF3B upregulated PTC-containing mRNAs while having minimal impact on the transcriptome in a hemophilia mouse model[21], which provides evidence that targeting branch-specific NMD factors could be both a safe and effective strategy for targeting nonsense mutations in the disease context. Importantly, the impact of inhibiting the SMG6-dependent NMD branch has not yet been evaluated in the lung. Previous studies suggest extensive redundancy exists between SMG6-mediated endonucleolytic and SMG5-SMG7-mediated exonucleolytic decay routes in regulating endogenous NMD substrates[18,19,35]. Therefore, inhibition of the SMG6-dependent branch may have less impact on endogenous NMD substrates while upregulating CFTR mRNAs with nonsense codons. However, further work is warranted to evaluate the safety of modulating the SMG6-dependent NMD branch in the lung under disease conditions.

## Methods

**Antisense oligonucleotide preparation.** Antisense oligonucleotides (ASOs) used in this study were 16-mer phosphorothioate oligonucleotides that contain constrained ethyl (cEt) modifications at positions 1–3 and 14–16, which flank a central 10-nucleotide deoxy gap. Oligonucleotides were synthesized using an Applied Biosystems 380B automated DNA synthesizer (PerkinElmer Life and Analytical Sciences, Richmond, CA) and purified as previously described[31]. Lyophilized ASOs are prepared for use by reconstitution in Dulbecco's buffered saline (Thermo Fisher Scientific, Carlsbad, CA) followed by filter sterilization using a 0.2-μm filter and quantification by UV spectrometry. ASO sequences will be provided upon request.

**Cell culture and ASO treatment.** 16HBE14o- cells, an immortalized human bronchial epithelial cell line[54], were genome-edited to produce CFF-16HBEge cell lines harboring CFTR Y122X, G542X, R1162X, or W1282X nonsense mutations as previously described[24]. CFF-16HBEge cell lines were provided by the Cystic Fibrosis Foundation. CFTR nonsense mutations in CFF-16HBEge cells exist within the endogenous CFTR allele and are expressed by the native CFTR promoter. CFF-16HBEge cells were cultured in MEM containing 10% fetal bovine serum (R&D Systems, Minneapolis, MN), 100 U/ml penicillin/ 100 U/ml streptomycin (Thermo Fisher Scientific, Carlsbad, CA), and Glutamax (Thermo Fisher Scientific, Carlsbad, CA) until 90% confluent. Cells were then seeded into collagen I-coated 96-well plates at a concentration of 10,000 cells/well and cultured until 90% confluent prior to ASO treatments. ASOs were diluted into fresh cell culture medium, which was subsequently added to the cells for treatment. Cells were treated for 6 days with replenishment of ASO-containing medium at day 3.

Primary human bronchial epithelial cells (hBEs) homozygous for the W1282X nonsense mutation were provided by the Cystic Fibrosis Foundation. hBE cells were thawed and cultured in Pneumacult EX expansion media (STEMCELL Technologies, Vancouver, Canada) in tissue culture flasks pre-coated with NIH3T3-conditioned medium. hBE cells were then seeded onto 0.4 μm pore size polycarbonate Transwell inserts (Corning, Corning, NY, #3381) pre-coated with NIH3T3-conditioned medium. hBE cells were then differentiated for >28 days at air–liquid interface in Pneumacult ALI medium (STEMCELL Technologies, Vancouver, Canada) with fresh

medium replenishment every 3 days[55]. hBE cells were treated with ASOs by free uptake from the basolateral compartment for a total of 6 days.

**RNA extraction and RT-qPCR.** Cultured CFF-16HBEge cells and hBE cells were lysed following ASO treatments and total RNA was prepared using the PureLink Total RNA Purification Kit as per the manufacturer's instructions (Invitrogen, Carlsbad, CA). The RT-qPCR analysis was performed with QuantStudio 6 and 7 Flex real-time PCR systems (Applied Biosystems, Foster City, CA) using AgPath-ID One-Step RT-PCR reagents (Applied Biosystems, Foster City, CA). RT-qPCR results were normalized to quantified total RNA using the Quant-iT Ribogreen RNA reagent (Molecular Probes, Eugene, OR) and to housekeeping gene β-actin. TaqMan primer-probe set sequences are provided (Supplementary Table 1).

**CFTR channel function analysis in CFF-16HBEge cells.** CFF-16HBEge cells were seeded onto collagen IV (Sigma-Aldrich, Cat# C7521, St. Louis, MO) pre-coated Snapwell inserts (Corning, Corning, NY; 12-mm filter diameter with 0.4-µm pore size) for 24 h at a density of $4.52 \times 10^5$ cells/cm². Cells were treated with ASOs at a final concentration of 1 µM by free uptake at liquid–liquid interface for a total of 6 days with treatment media replenished at day 3, with or without 100 µM geneticin (G418) (Thermo Fisher Scientific, Carlsbad, CA) added for the final 48 h. Snapwell inserts were then mounted into Ussing chambers and short circuit current (Isc) measurements were recorded with the VCC MC8 multichannel voltage-current clamp amplifier (Physiologic Instruments, San Diego, CA). Cells were bathed in a basolateral to apical chloride ion gradient. Composition for basolateral buffer was 137 mM NaCl, 4 mM KCl, 1.8 mM CaCl₂, 1 mM MgCl₂, 10 mM HEPES, 10 mM D-(+) glucose (Sigma-Aldrich, St. Louis, MO), and apical buffer composition was 137 mM sodium-gluconate, 4 mM KCl, 1.8 mM CaCl₂, 1 mM MgCl₂, 10 mM HEPES, and 10 mM D-(+)-glucose (Sigma-Aldrich, St. Louis, MO). All buffers were adjusted to physiological pH 7.4. Isc and epithelial resistance were calculated using Acquire & Analyze software (Physiologic Instruments, San Diego, CA). To assess CFTR-mediated current, the following compounds were added sequentially: 7 µM benzamil (Selleckchem, Houston, TX) to the apical side, 20 µM forskolin (Sigma-Aldrich, St. Louis, MO) to apical and basolateral sides, 1 µM VX-770 (Selleckchem, Houston, TX) to the apical side, and 20 µM CFTR inhibitor-172 (Selleckchem, Houston, TX) to the apical side. The current (Isc) mediated by CFTR was calculated as the difference (ΔIsc) from stabilized current achieved after forskolin addition or VX-770 addition following the baseline achieved by addition of CFTR inhibitor-172 (CFTRinh-172)-mediated current stabilization.

**Immunoblotting.** CFTR and NMD factor protein levels were measured by western blot analysis. Cell protein lysate was collected with IP Lysis Buffer containing Halt protease and phosphatase inhibitor cocktail (Thermo Fisher Scientific, Carlsbad, CA). Protein concentration was determined by the DC protein assay (Bio-Rad, Hercules, CA). A total of 40–50 µg of protein was separated by gel electrophoresis using a 3–8% Criterion XT Tris-acetate protein gel and transferred to a 0.2-µm PVDF membrane (Bio-Rad, Hercules, CA). Membranes were probed with mouse IgG2β anti-human CFTR (UNC596; 1:2000 dilution, UNC CFTR Antibody Distribution Program, Chapel Hill, NC) followed by goat anti-mouse HRP-conjugated secondary antibody (Thermo Fisher Scientific, #31430, 1:10000 dilution, Carlsbad, CA) and detected with SuperSignal West Femto Maximum Sensitivity Substrate (Thermo Fisher Scientific, Carlsbad, CA). Membranes were then probed with anti-Na⁺/K⁺-ATPase (Santa Cruz Biotechnology, #sc-21712, 1:20,000 dilution, Dallas, TX) followed by detection with ECL Prime detection reagent (GE Healthcare; Chicago, IL). Signals following probing for anti-β-actin (Cell Signaling Technologies, #3700, 1:2000 dilution, Danvers, MA), anti-SMG6 (Abcam, #ab87539, 1:2000, Cambridge, UK), and anti-SMG1 (Bethyl Laboratories, #A300-393A, 1:2000 dilution, Montgomery, TX) antibodies were assessed using the Odyssey infrared system (Li-COR, Lincoln, NE) following incubation with IRDye anti-mouse (Li-COR, 1:5000 dilution, #926-68070, Lincoln, NE) and IRDye anti-rabbit (Li-COR, 1:5000 dilution, #926-32211, Lincoln, NE) secondary antibodies. Protein expression was quantified using either ImageJ or Li-COR Odyssey software. Unless otherwise noted, only the mature, fully-glycosylated C band of CFTR protein was quantified.

**Statistical analysis.** GraphPad Prism version 7.01 software (GrapPad Prism, Inc., San Diego, CA) was used to perform statistical analyses. Data are presented as mean ± SEM and are representative of at least two independent experiments performed in triplicate. One-way or two-way ANOVA was performed for statistical analyses between three or more groups followed by the indicated multiple-comparison tests. $p < 0.05$ was considered statistically significant. Exact p-values for each indicated comparison are provided in the Source Data file.

**Reporting summary.** Further information on research design is available in the Nature Research Reporting Summary linked to this article.

## Data availability

The data supporting the findings of this study are available from the corresponding authors upon reasonable request. Source data for the figures and supplementary figures are provided as a Source Data file. Source data are provided with this paper.

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

## Acknowledgements

This work was supported by Ionis Pharmaceuticals and by a Cystic Fibrosis Foundation award to L.H. (IONIS18X0-SC). The authors would like to thank Yi Cheng, Hillary Valley, Kevin Coote, Jan Harrington, and Hermann Bihler from the Cystic Fibrosis Foundation for providing the CFF-16HBEge cells, hBE cells, and helpful advice related to cell culture and electrophysiology applications. We would also like to thank Tracy Reigle, Michael Cichanowski, and Wanda Sullivan for their help preparing the figures.

## Author contributions

E.J.S, M.M.K, and L.H designed and executed experiments and performed data analysis. E.J.S, M.M.K, M.M, A.S.R, B.P.M, S.G, and L.H conceptualized experimental approaches and interpreted data. L.H and E.J.S wrote the manuscript. All authors reviewed and edited the manuscript.

## Competing interests

E.J.S, M.M.K, A.S.R, S.G, L.H, and B.P.M are employees and shareholders of Ionis Pharmaceuticals. M.M is an employee of the Cystic Fibrosis Foundation.
