## [Peer Review File · Nature Communications]

Title: CFTR mRNAs with nonsense codons are degraded by the SMG6-mediated endonucleolytic pathwayREVIEWER COMMENTS

Reviewer #1 (Remarks to the Author):

Sanderlin and authors have shown using a panel of nonsense-mediated decay (NMD) factor targeting ASOs, that CFTR mRNAs which harbor common CF nonsense mutations are regulated by different NMD pathways. Using gene edited human bronchial epithelial cell lines with Y122X-, G542X-, R1162X- and W1282X-CFTR CF mutations, it was found that the Y122X mRNA specifically was not regulated by UPF2, UPF3 and exon joining complexes. However, all CFTR mRNA NMD was regulated by SMG1 and SMG6, where ASO targeting resulted in near WT mRNA expression. This study presents clear results that not all PTCs within a mRNA are regulated by NMD in the same fashion. Understanding which NMD pathways regulate CFTR mRNA with nonsense mutations is not only important for our basic understanding of NMD biology, but also for the development of nonsense-associated disease therapeutics. Here it is speculated, that discrepancies in mRNA NMD pathways may be exploited for therapeutic development.

The data are of high quality, the results are intriguing, and the implications are of high importance to the field. The manuscript would be strengthened by the authors addressing the following issues.

Issues to Address

- 1) For the general audience, it would be beneficial to include a figure with a schematic of the NMD branches and the factors that were targeted in this study.
- 2) In all qRT-PCR experiments, the expression of CFTR mRNA is normalized to total RNA. Because the authors are targeting NMD pathways, it should be determined that total RNA expression is not significantly changing with ASO treatment.
- 3) It should be made clearer that the results in Figure 5E are obtained from primary HBE cells and not 16HBEge cells in the figure itself.
- 4) The authors are thorough throughout the study in comparing CFTR mRNA expression results from 16HBEge cell lines treated with various ASOs to the parental 16HBEo- cells line. However, this comparison is not made with Ussing chamber recordings of CFTR short-circuit current in Figure 7. Ultimately, the percent rescue of CFTR channel function is what is important for development of a CF therapeutic. It would be helpful if %WT short-circuit current values were included as a right Y-axis in Figure 7 B, D, F, H. While it is understood that this comparison is maybe not as accurate to perform with WB detection of CFTR protein, it would also be helpful to include this information in the results section.
- 5) The concentration of G418 should be included in the main text and Figure 7 legend. Furthermore, at the concentration of G418 used in this study (100uM), general translation fidelity is significantly affected (PMID: 31971508) and WT CFTR protein expression and function are significantly reduced in 16HBE14o-cells (PMID: 30563749). These findings should be discussed. Some discussion should be included to underscore the challenge of finding non-toxic yet potent PTC readthrough agents to use in combination with NMD targeting ASOs.

6) While the source of the 16HBE14o- and 16HBEge cells were cited, a better introduction of the cells and why they were used in this study is needed.

7) The authors discuss the possibility of targeting specific NMD pathways as an avenue for PTC therapeutics. It is understood that being able to target specific branches of NMD should be less toxic. However, the authors should add some discussion about the possible toxic side effects NMD inhibition that includes the possible number of endogenous targets of SMG6 endonuclease and why targeting SMG6 over SMG1 or SMG5/SMG7 pathways would be beneficial.

8) It is peculiar that knockdown of SMG1 resulted in significant “super rescue” of G542X-CFTR mRNA expression (Fig 3), while other mutations exhibited near WT mRNA expression. It may be expected that with SMG1 ASO/G418 treatment, G542X would have the highest CFTR protein and functional rescue, but it does not. Do the authors have a possible mechanism behind this? There are general inconsistencies between mRNA, protein and functional rescue with the different CFTR PTCs and ASO/G418 treatment. Is this because of the G418 induced near cognate suppression and the different sensitivities of CFTR PTCs to amino acid incorporation into CFTR protein?

Reviewer #2 (Remarks to the Author):

General comments:

Nonsense mutations can result in a significant reduction of gene expression partially due to rapid mRNA degradation through the NMD pathway. Recent studies have reported there are three branches of the NMD pathway diverging at the stage of PTC recognition including UPF2-independent, UPF3B-independent, and EJC-independent branches, however, it has not been thoroughly investigated which branch of the NMD pathway governs the decay of CFTR mRNAs containing nonsense codons.

In this well-written paper, Dr. Sanderlin, et al, used ASOs to reduce the expression of each NMD pathway component to determine which NMD branch regulates the degradation of CFTR mRNAs containing nonsense codons. The authors have provided convincing data to confirm that the core NMD factors UPF1 and SMG1 regulate all evaluated CFTR mRNAs with nonsense codons, the CFTR mRNAs with G542X, R1162X, and W1282X nonsense codons require UPF2, UPF3, and exon junction complex proteins for NMD, whereas CFTR mRNAs with the Y122X nonsense codon do not. They demonstrated that all evaluated CFTR mRNAs harboring nonsense codons were degraded by the SMG6-mediated endonucleolytic pathway rather than the SMG5/SMG7-mediated exonucleolytic pathway, this novel finding suggests inhibition of the branch-specific NMD factor SMG6 is a promising therapeutic target for the stabilization of CFTR mRNAs harboring prevalent nonsense codons. They also demonstrated that stabilization of CFTR mRNAs by NMD inhibition alone improved functional W1282X protein production, and improved the efficiency of aminoglycoside translational readthrough of CFTR-Y122X, -G542X, and -R1162X mRNAs.

Overall, the findings are justified and substantiated, and will be of interest to the field, particularly

regarding propensity to augment readthrough agents, although in some cases additional data are needed or wording of conclusions modified, as outlined below.

Major critiques:

1. Use of primary HBE cells is used only for the very selective W1282X condition. This has been adopted as the standard assessment to ascertain clinical relevance, and should be included for some of the key mutations studies where feasible.

2. Negative scrambled ASO controls are needed for several figures (instead of only PBS control), as they are used inconsistently. This is important as effect on cell membranes are possible with nucleotide therapy that could influence expression or ion transport function, especially with electrochemical gradients.

3. About the term “stabilization/stabilizing CFTR mRNA”

The stabilization/stabilizing CFTR mRNAs means the CFTR mRNAs were maintained at a certain level which could be high or low levels. In this paper, the inhibition of certain NMD factors (UPF1, SMG1, UPF2, SMG6) by ASOs causes upregulation of CFTR mRNAs rather than just stabilization compared to the control groups (PBS or control ASO). Can the authors explain this effect? It might be better to use “upregulation/upregulating CFTR mRNAs”. In many cases, since mRNA half life is not estimated, only steady state levels, wording should be adjusted.

4. About RT-qPCR normalization:

In the paper, RT-qPCR results were normalized to quantified total RNA. It will be ok if the same amount of RNA with the same quality was used in the RT-qPCR. Do all RNA samples have the same quality? The OD260/OD280 will give the information about the RNA purity, and the RIN (RNA Integrity Number) value will give the information about the RNA quality. The RNAs will be good for downstream application if a RIN is higher than five as good total RNA quality and higher than eight as perfect total RNA. (References: Schroeder A, et al. BMC Mol Biol. 2006; 7: 3, Mol Aspects Med. 2006; 27(2-3):126-39). If the authors did not measure the RIN, it may be better to normalize the target gene expression to the housekeeping gene expression.

5. Figure 6:

Figure 6 does not convincingly discriminate between the WT band migration the truncated band migration – migration of the C band looks similar to R1162X, for example, yet conclusions are different for those to genotypes. If confirmed, truncated vs. non truncated band should be designated on the quantitative results.

On a related point, in the methods for the Immunoblotting, row 380-381 on page 17, the authors claimed that “Only the mature, fully-glycosylated C band of CFTR protein was quantified.” However, the upper bands in Figure 6G are the truncated CFTR protein rather than full-length CFTR protein, so authors need to clarify this and revise the legend for Figure 6H.

Figure 6 would also benefit from semi quantitative normalization to WT CFTR protein levels shown on the blot.

6. Figure 1: 150 uA/cm² is unusually high current values for WT parental 16HBE. That raises concern that the currents observed are disproportionately high bar for mutation specific assessments. Are the particular aspects to the conditions used that are bringing about those currents.

7. About the effect of different combinations of ASOs on the CFTR mRNAs with nonsense mutations
The effect of the SMG8+SMG9 ASOs (Fig.3), UPF3A+UPF3B ASOs (Fig.4), and SMG5+SMG7 ASOs (Fig.5) on CFTR mRNAs with nonsense mutations were tested. However, the combo of the core NMD factor SMG1 and the branch-specific NMD factor SMG6 ASOs was not tested. They may have a synergistic effect on stabilizing/upregulating the CFTR mRNAs which could result in more production of the functional full length CFTR protein. It could also reveal if more than one factor is relevant to the NMD process by these pathways, an important aspect of characterizing the relevant branch.

Minor comments:

1. Page 5, row 105-108 and Supplementary Fig.1:

The text claimed “We were unable to detect full-length CFTR proteins in any of these cell lines by western blot analysis (Fig. 1B). However, the presence of the partially stable, C-truncated W1282X-CFTR protein was detected (Supplementary Fig. 1), as previously characterized 25, 27-30.” But the legend for Supplementary Figure 1 claimed that “CFF-16HBEge cells harboring Y122X or W1282X CFTR nonsense mutations have detectible CFTR truncated protein products. Qualitative western blot analysis for C-truncated W1282X CFTR proteins in CFF-16HBEge-W1282X cells. Na⁺/K⁺-ATPase (NKA) was used as a loading control.” It is quite possible the truncated CFTR protein was detected only in the 16HBE W1282X cells rather than in the Y122X cells.

2. Page 6, and Page 27 about Figure 2:

It would be easier for readers to compare the mRNA change of the same NMD factor after different ASOs treatment if the NMD factors on the X-axis were labeled at the same order. For example, SMG1. SMG5, SMG6, SMG7, SMG8, SMG9, UPF1, UPF2, UPF3A, UPF3B.

The Y-axis on Figure 2 I should be Relative mRNA Levels.

3. Page 29-34, about Figure 3, 4, and 5

In the legend “Wild-type WT CFTR mRNA levels in parental cells are indicated by orange dotted lines,” but the “orange” lines in all figures look like “red”.

4. To justify conclusions regarding Figure 1D, comparisons are needed between Y122X and other mutations.

Reviewer #3 (Remarks to the Author):

In this study, the authors showed that (1) CFTR mRNAs with several nonsense mutations are degraded by the SMG6-dependent, as opposed to the SMG5/7-dependent, pathway, (2) gapmer ASOs that target key NMD factors, including SMG6, can restore the levels of CFTR mRNAs with the nonsense mutations, and in combination with a readthrough drug, the ASOs can restore the CFTR function to some extent, (3)

one of the tested nonsense mutations (Y122X), while requiring UPF1 and SMG6 for NMD as other CFTR nonsense mutations, does not require UPF2, UPF3, or exon junction complex formation. While it is a well-executed study with clear and useful conclusions, I am not fully convinced that the impact and clinical implications of the conclusions merit publication in Nature Communications. My opinions are summarized below.

Impact of the main conclusion: The scientific value of the main finding — all the tested CFTR nonsense mutations are SMG6-dependent — is not clear. How often NMD is mediated by the SMG6 pathway, as opposed to the SMG5/7 pathway? Is it 10% or 90%? If it is the former, the finding would have had greater scientific implications because readers, if it wasn't for this study, would have assumed that they are dependent on the SMG5/7 pathway. Such conclusion would also have greater clinical implications too because, if the SMG6 pathway controls only a small fraction of genes, therapeutically knocking down the pathway would have smaller side effects. However, recent studies have suggested that the SMG6-mediated endonucleolytic cleavage pathway is likely a more prevalent mechanism of NMD than the SMG5/7-mediated pathway, as the loss of SMG6 impaired NMD more severely than the loss of SMG7 (Boehm et al., 2021 Nat Comm). If the SMG6 pathway is the main pathway anyway, the fact that CFTR nonsense mutations are dependent on SMG6 have only limited impact.

Therapeutic potential: The authors mentioned that the SMG6-targeting gapmer ASOs could be a therapeutic strategy for CF. While it is agreeable that targeting a branch-specific NMD factor, such as SMG6, would have less side effects than targeting universal factors, such as UPF1 or SMG1. However, targeting SMG6 would still lead to changes in expression of a sizable number of genes (it has not been mentioned in the manuscript how many normal genes are differentially regulated by SMG6 knock-down). Therefore, therapeutic use of SMG6 gapmer ASOs can lead to significant side effects, and unless the authors experimentally demonstrate the safety of the gapmer ASOs (which is not given in the manuscript), it is hard to justify the claim that the gapmer ASOs are suitable drug candidates. This is a key issue in terms of the impact of this study, because if the gapmer ASOs were to have questionable therapeutic implications, a relatively simpler study using siRNAs for the NMD factors would have led to the same conclusions as the current study. (siRNA experiments are simpler because pre-designed and knock-down guaranteed siRNAs for each gene are readily commercially available off the shelf.) Another point to consider when assessing the clinical impact of this study is that ASO approaches targeting exon junction complex formation of a specific target gene to inhibit NMD of a specific target gene have been published. This alternative approach seems conceptually superior to the approach described in this study because of the specificity, hence potentially lower side effects. Another issue that makes it difficult to assess the clinical significance of this work is that in Fig. 7, the authors showed that the ASOs, when used in combination with a readthrough drug, can restore the CFTR function to some extent. However, since no positive control (i.e., CFTR carrier cells) was performed, it is hard to assess whether or not the improved CFTR function is going to provide meaningful clinical improvement.

Mutation selection: When the authors could have chosen any nonsense mutations of CFTR (mutation-harboring cell lines can be generated by CRISPR knock-in as done in this study, or for common mutations, patient-derived cell lines can be obtained from the foundation), they chose CFTR Y122X,

G542X, R1162X, or W1282X. An issue here is that they did not explain the selection criteria. While G542X and W1282X are reasonable choices in that they are high frequency mutations (2.5% and 1.2% allele frequency in CF patients, https://cftr2.org/mutations_history), Y122X has a very low frequency (0.06%). It is also not explained why R553X is not chosen in this study, when it has a 10 times higher frequency (0.9%) than Y122X. This study highlights Y122X mutation because is degraded by a different NMD pathway compared to other CFTR nonsense mutations, but it has not been mentioned that the clinical importance of this mutation is limited as it has a low allele frequency in patients. In addition, the authors did not provide any mechanistic basis for the UPF2/3-independent decay of Y122X transcripts (i.e., what are the determinants of pathway selection).

REVIEWER COMMENTS

Reviewer #1 (Remarks to the Author):

Sanderlin and authors have shown using a panel of nonsense-mediated decay (NMD) factor targeting ASOs, that CFTR mRNAs which harbor common CF nonsense mutations are regulated by different NMD pathways. Using gene edited human bronchial epithelial cell lines with Y122X-, G542X-, R1162X- and W1282X-CFTR CF mutations, it was found that the Y122X mRNA specifically was not regulated by UPF2, UPF3 and exon joining complexes. However, all CFTR mRNA NMD was regulated by SMG1 and SMG6, where ASO targeting resulted in near WT mRNA expression. This study presents clear results that not all PTCs within a mRNA are regulated by NMD in the same fashion. Understanding which NMD pathways regulate CFTR mRNA with nonsense mutations is not only important for our basic understanding of NMD biology, but also for the development of nonsense-associated disease therapeutics. Here it is speculated, that discrepancies in mRNA NMD pathways may be exploited for therapeutic development.

The data are of high quality, the results are intriguing, and the implications are of high importance to the field. The manuscript would be strengthened by the authors addressing the following issues.

Response: We thank this reviewer for the careful consideration of our manuscript. Please find our point-by-point responses to the comments below. We believe that the suggested changes improved the manuscript.

Issues to Address

1) For the general audience, it would be beneficial to include a figure with a schematic of the NMD branches and the factors that were targeted in this study.

Response: Thank you for this suggestion. We added a schematic of the NMD pathway as a new Figure 1F in the revised manuscript including the core and branch-specific NMD factors targeted by ASOs.

2) In all qRT-PCR experiments, the expression of CFTR mRNA is normalized to total RNA. Because the authors are targeting NMD pathways, it should be determined that total RNA expression is not significantly changing with ASO treatment.

Response: Thank you for this comment. We reanalyzed all RT-qPCR results using housekeeping gene β -actin mRNA as an endogenous control for normalization. The new results are consistent with our original results obtained using the total RNA amount for normalization. We added these new analyses as Supplemental Figure 2, 3, 5 and 7B in addition to the original analyses.

3) It should be made clearer that the results in Figure 5E are obtained from primary HBE cells and not 16HBEge cells in the figure itself.

Response: Thank you for this suggestion. We have added "Primary hBE cells (W1282X/W1282X)" to Figure 5E title.

4) The authors are thorough throughout the study in comparing CFTR mRNA expression results from 16HBEge cell lines treated with various ASOs to the parental 16HBEo- cells line. However, this comparison is not made with Ussing chamber recordings of CFTR short-circuit current in Figure 7. Ultimately, the percent rescue of CFTR channel function is what is important for development of a CF therapeutic. It would be helpful if %WT short-circuit current values were included as a right Y-axis in Figure 7 B, D, F, H. While it is understood that this comparison is maybe not as accurate to perform with WB detection of CFTR protein, it would also be helpful to include this information in the results section.

Response: Thank you for this suggestion. We included a right Y-axis in Figure 7 B, D, F, and H which provides a comparison of CFTR channel function in 16HBEge cells carrying CFTR nonsense alleles with WT-CFTR channel function from 16HBE14o- parental cells.

Although we agree that a calculation of CFTR protein in 16HBEge cells to the percentage of WT CFTR protein in parental cells would be helpful, we share the concern with this reviewer that the calculation may not be accurate enough to provide meaningful information due to the low abundance of the CFTR proteins in 16HBEge cells and the semi-quantitative nature of the western blot assay. As such, we decided not to include this calculation in our manuscript.

5) The concentration of G418 should be included in the main text and Figure 7 legend. Furthermore, at the concentration of G418 used in this study (100uM), general translation fidelity is significantly affected (PMID: 31971508) and WT CFTR protein expression and function are significantly reduced in 16HBE14o-cells (PMID: 30563749). These findings should be discussed. Some discussion should be included to underscore the challenge of finding non-toxic yet potent PTC readthrough agents to use in combination with NMD targeting ASOs.

Response: Thank you for this suggestion. We added the concentration used of G418 in the main text of the results section (line 219) and in the figure legends of Figures 6-7 and Supplementary Figures 8-11. We have cited the provided literature and included additional discussion of the challenge of finding efficacious and safe PTC readthrough agents (Line 307-318).

6) While the source of the 16HBE14o- and 16HBEge cells were cited, a better introduction of the cells and why they were used in this study is needed.

Response: We have added a more thorough introduction to the CFF-16HBEge cells (line 97-108).

7) The authors discuss the possibility of targeting specific NMD pathways as an avenue for PTC therapeutics. It is understood that being able to target specific branches of NMD should be less toxic. However, the authors should add some discussion about the possible toxic side effects NMD inhibition that includes the possible number of endogenous targets of SMG6 endonuclease and why targeting SMG6 over SMG1 or SMG5/SMG7 pathways would be beneficial.

Response: Thank you for the thoughtful comment. We added additional discussion points related to the opportunity of targeting branch-specific factors such as SMG6, which may have less impact on the transcriptome when compared to core NMD factors (line 328-334). We have also added discussion related to the possible impacts on the transcriptome for targeting branch-specific NMD factor SMG6 and the need for further evaluation of the safety of targeting branch-specific NMD factors in the lung.

8) It is peculiar that knockdown of SMG1 resulted in significant “super rescue” of G542X-CFTR mRNA expression (Fig 3), while other mutations exhibited near WT mRNA expression. It may be expected that with SMG1 ASO/G418 treatment, G542X would have the highest CFTR protein and functional rescue, but it does not. Do the authors have a possible mechanism behind this? There are general inconsistencies between mRNA, protein and functional rescue with the different CFTR PTCs and ASO/G418 treatment. Is this because of the G418 induced near cognate suppression and the different sensitivities of CFTR PTCs to amino acid incorporation into CFTR protein?

Response: We agree these are interesting observations as to the non-linear relationship in magnitude of CFTR protein and function with respect to mRNA levels following NMD inhibition and translational readthrough induction by G418. As suggested by the reviewer, PTC identity, sequence context of the PTC, and the identity of the incorporated near cognate-tRNA have all been indicated in literature to affect translational readthrough efficiency, protein stability, and protein function. We added a discussion point in the revised manuscript to provide a possible mechanism for this observation (Line 304-311).

Reviewer #2 (Remarks to the Author):

General comments:

Nonsense mutations can result in a significant reduction of gene expression partially due to rapid mRNA degradation through the NMD pathway. Recent studies have reported there are three branches of the NMD pathway diverging at the stage of PTC recognition including UPF2-independent, UPF3B-independent, and EJC-independent branches, however, It has not been thoroughly investigated which branch of the NMD pathway governs the decay of CFTR mRNAs containing nonsense codons. In this well-written paper, Dr. Sanderlin, et al, used ASOs to reduce the expression of each NMD pathway component to determine which NMD branch regulates the degradation of CFTR mRNAs containing nonsense codons. The authors have provided convincing data to confirm that the core NMD factors UPF1 and SMG1 regulate all evaluated CFTR mRNAs with nonsense codons, the CFTR mRNAs with G542X, R1162X, and W1282X nonsense codons require UPF2, UPF3, and exon junction complex proteins for NMD, whereas CFTR mRNAs with the Y122X nonsense codon do not. They demonstrated that all evaluated CFTR mRNAs harboring nonsense codons were degraded by the SMG6-mediated endonucleolytic pathway rather than the SMG5/SMG7-mediated exonucleolytic pathway, this novel finding suggests inhibition of the branch-specific NMD factor SMG6 is a promising therapeutic target for the stabilization of CFTR mRNAs harboring prevalent nonsense codons. They also demonstrated that stabilization of CFTR mRNAs by NMD inhibition alone improved functional W1282X protein production, and improved the efficiency of aminoglycoside translational readthrough of CFTR-Y122X, -G542X, and -R1162X mRNAs.

Overall, the findings are justified and substantiated, and will be of interest to the field, particularly regarding propensity to augment readthrough agents, although in some cases additional data are needed or wording of conclusions modified, as outlined below.

Response: We thank this reviewer for the thoughtful consideration of our findings in this manuscript. Please find our point-by-point responses to the comments below. We believe that the suggested changes improved the manuscript.

Major critiques:

1. Use of primary HBE cells is used only for the very selective W1282X condition. This has been adopted as the standard assessment to ascertain clinical relevance, and should be included for some of the key mutations studies where feasible.

Response: We agree that evaluation of additional CFTR nonsense mutations in primary hBE cells would be of value for this study. However, at the time of this study, it was not feasible to obtain either hBE or hNE (human primary nasal epithelial) cells with other CFTR nonsense mutations due to limited availability. The availability of hBE and hNE cells with CFTR nonsense mutations is a common obstacle for the CF research community and is in part the motivation for the development of the CFF-16HBE cell model system.

2. Negative scrambled ASO controls are needed for several figures (instead of only PBS control), as they are used inconsistently. This is important as effect on cell membranes are possible with nucleotide therapy that could influence expression or ion transport function, especially with electrochemical gradients.

Response: Thank you for highlighting this inconsistency in the data presentation. In the original manuscript, we did not provide negative control ASO results for Figure 2, which demonstrates ASO-mediated reduction of the NMD factors. We included negative control ASO data for Figure 2 in the revised manuscript as Supplemental Figure 3.

3. About the term “stabilization/stabilizing CFTR mRNA”

The stabilization/stabilizing CFTR mRNAs means the CFTR mRNAs were maintained at a certain level which could be high or low levels. In this paper, the inhibition of certain NMD factors (UPF1, SMG1, UPF2, SMG6) by ASOs causes upregulation of CFTR mRNAs rather than just stabilization compared to the control groups (PBS or control ASO). Can the authors explain this effect? It might be better to use “upregulation/upregulating CFTR mRNAs”. In many cases, since mRNA half life is not estimated, only steady state levels, wording should be adjusted.

Response: Thank you for this suggestion. We changed the wording “stabilization/stabilize” to “upregulation/upregulate” to reflect the observed upregulation of CFTR transcripts throughout the manuscript.

4. About RT-qPCR normalization:

In the paper, RT-qPCR results were normalized to quantified total RNA. It will be ok if the same amount of RNA with the same quality was used in the RT-qPCR. Do all RNA samples have the same quality? The OD260/OD280 will give the information about the RNA purity, and the RIN (RNA Integrity Number) value will give the information about the RNA quality. The RNAs will be good for downstream application if a RIN is higher than five as good total RNA quality and higher than eight as perfect total RNA. (References: Schroeder A, et al. BMC Mol Biol. 2006; 7: 3, Mol Aspects Med. 2006; 27(2-3):126-39). If the authors did not measure the RIN, it may be better to normalize the target gene expression to the housekeeping gene expression.

Response: Thank you for providing this helpful comment. We evaluated RNA quality using the Agilent 2100 Bioanalyzer from pilot studies and found RNA Integrity Numbers above 9.5 irrespective of treatment condition in 16HBE cells. These pilot experiments suggested that the methodology used for RNA collection yields high quality RNA for all RT-qPCR experiments. Furthermore, to address this concern, we reanalyzed all RT-qPCR results for NMD factor and CFTR mRNA levels using housekeeping gene β -actin mRNA as an endogenous control for normalization. The new analyses are consistent with our original results obtained using the total RNA amount for normalization. We added these new analyses as Supplemental Figure 2, 3, 5 and 7B in addition to the original analyses.

5. Figure 6:

Figure 6 does not convincingly discriminate between the WT band migration the truncated band migration – migration of the C band looks similar to R1162X, for example, yet conclusions are different for those to genotypes. If confirmed, truncated vs. non truncated band should be designated on the quantitative results.

On a related point, in the methods for the Immunoblotting, row 380-381 on page 17, the authors claimed that “Only the mature, fully-glycosylated C band of CFTR protein was quantified.” However, the upper bands in Figure 6G are the truncated CFTR protein rather than full-length CFTR protein, so authors need to clarify this and revise the legend for Figure 6H.

Figure 6 would also benefit from semi quantitative normalization to WT CFTR protein levels shown on the blot.

Response: Thank you for pointing out these inconsistencies. We have modified the title of Figure 6H to indicate that the C-truncated CFTR-W1282X protein product was quantified. We have also revised the Figure 6 legend to indicate full-length CFTR protein was quantified from CFF-16HBEge-Y122X, G542X, and R1162X cells and C-truncated CFTR-W1282X protein was quantified for 16HBEge-W1282X cells. We also agree that normalization of CFTR protein upregulation in 16HBEge cells to WT CFTR protein levels would be of benefit to include, but have concern the estimation of the percentage of WT CFTR protein levels may not be accurate enough to provide meaningful interpretation in this study. Our concerns are related to the low CFTR protein levels in 16HBEge cells relative to WT CFTR in parental cells for semi-quantitative western blot analysis. As such, we have decided not to include this comparison information for the western blot analysis of CFTR protein.

6. Figure 1: 150 $\mu\text{A}/\text{cm}^2$ is unusually high current values for WT parental 16HBE. That raises concern that the currents observed are disproportionately high bar for mutation specific assessments. Are the particular aspects to the conditions used that are bringing about those currents.

Response: We share the concern raised by this reviewer that the WT CFTR function from 16HBE14o- cells may present a high bar for comparison to 16HBEge cells which could underrepresent the efficacy of the treatment strategies in this study. Please note in Fig. 1D, we did not subtract baseline currents when plotting the representative I_{sc} tracings. The ΔI_{sc} of WT CFTR function in parental cells is $\sim 118 \mu\text{A}/\text{cm}^2$, as presented in Fig. 1E, which is within the normal range. The 16HBE14o- parental cell I_{sc} data presented in Figure 1D and 1E are obtained from more than four independent experiments employing consistent conditions between all experiments, such as the establishment of a basolateral to apical chloride ion gradient.

7. About the effect of different combinations of ASOs on the CFTR mRNAs with nonsense mutations
The effect of the SMG8+SMG9 ASOs (Fig.3), UPF3A+UPF3B ASOs (Fig.4), and SMG5+SMG7 ASOs (Fig.5) on CFTR mRNAs with nonsense mutations were tested. However, the combo of the core NMD factor SMG1 and the branch-specific NMD factor SMG6 ASOs was not tested. They may have a synergistic effect on stabilizing/upregulating the CFTR mRNAs which could result in more production of the functional full length CFTR protein. It could also reveal if more than one factor is relevant to the NMD process by these pathways, an important aspect of characterizing the relevant branch.

Response: Thank you for this suggestion. We agree that any further upregulation of CFTR mRNAs would be beneficial for potential subsequent increases of CFTR protein and function. However, we believe that combining ASOs targeting both core NMD factors (e.g. SMG1 or UPF1) and branch-specific NMD factors will have too large of an impact on the normal transcriptome and less likely to be a feasible therapeutic approach. Therefore, we did not test these combinations in this study. Our rationale for combining ASOs targeting branch-specific NMD factors UPF3A + UPF3B ASOs is due to the functional redundancy of each NMD factor which regulates smaller pools of NMD substrates. We also combined ASOs targeting SMG8+SMG9 or branch-specific factors SMG5+SMG7 as these factors heterodimerize for NMD pathway effector functions.

Minor comments:

1. Page 5, row 105-108 and Supplementary Fig.1:

The text claimed “We were unable to detect full-length CFTR proteins in any of these cell lines by western blot analysis (Fig. 1B). However, the presence of the partially stable, C-truncated W1282X-CFTR protein was detected (Supplementary Fig. 1), as previously characterized 25, 27-30.” But the legend for Supplementary Figure 1 claimed that “CFF-16HBEge cells harboring Y122X or W1282X CFTR nonsense mutations have detectible CFTR truncated protein products. Qualitative western blot analysis for C-truncated W1282X CFTR proteins in CFF-16HBEge-W1282X cells. Na⁺/K⁺-ATPase (NKA) was used as a loading control.” It is quite possible the truncated CFTR protein was detected only in the 16HBE W1282X cells rather than in the Y122X cells.

Response: Thank you for identifying this error written in the figure legend for Supplementary Figure 1 indicating the identification of truncated CFTR protein products in CFF-16HBEge-Y122X cells. We deleted this incorrect statement to reflect the western blot image of C-truncated CFTR protein in 16HBEge-W1282X cells in the revised manuscript.

2. Page 6, and Page 27 about Figure 2:

It would be easier for readers to compare the mRNA change of the same NMD factor after different ASOs treatment if the NMD factors on the X-axis were labeled at the same order. For example, SMG1, SMG5, SMG6, SMG7, SMG8, SMG9, UPF1, UPF2, UPF3A, UPF3B.

The Y-axis on Figure 2 I should be Relative mRNA Levels.

Response: Thank you for this suggestion. We adjusted the orders of the NMD factors on the X-axis as: UPF1, UPF2, UPF3A, UPF3B, SMG1, SMG5, SMG6, SMG7, SMG8, and SMG9. We also rearranged Fig. 2 panels to present ASO treatments in the same order. Y-axis in Figure 2I was changed to “Relative mRNA levels”.

3. Page 29-34, about Figure 3, 4, and 5

In the legend “Wild-type WT CFTR mRNA levels in parental cells are indicated by orange dotted lines,” but the “orange” lines in all figures look like “red”.

Response: We changed the legends for Figures 3-5 to include a more accurate description of the colored dotted line from orange to red for the WT CFTR mRNA levels.

4. To justify conclusions regarding Figure 1D, comparisons are needed between Y122X and other mutations.

Response: Thank you for this suggestion. We performed comparisons of basal CFTR function between CFF-16HBEge-Y122X and other cell lines. The results demonstrate that 16HBEge-Y122X cells have significantly higher basal CFTR function compared to other 16HBEge cells. We added this analysis to Figure 1D.

Reviewer #3 (Remarks to the Author):

In this study, the authors showed that (1) CFTR mRNAs with several nonsense mutations are degraded by the SMG6-dependent, as opposed to the SMG5/7-dependent, pathway, (2) gapmer ASOs that target key NMD factors, including SMG6, can restore the levels of CFTR mRNAs with the nonsense mutations, and in combination with a readthrough drug, the ASOs can restore the CFTR function to some extent, (3) one of the tested nonsense mutations (Y122X), while requiring UPF1 and SMG6 for NMD as other CFTR nonsense mutations, does not require UPF2, UPF3, or exon junction complex formation. While it is a well-executed study with clear and useful conclusions, I am not fully convinced that the impact and clinical implications of the conclusions merit publication in Nature Communications. My opinions are summarized below.

Response: We thank this reviewer for carefully reviewing our work. However, we respectfully disagree with the reviewer’s opinion on the impact of our manuscript. This manuscript is the first study identifying which branch of the NMD pathway degrades CFTR mRNAs containing nonsense codons and whether the position of the nonsense codon affects branch specificity. We believe this work also provides the foundation needed for developing therapies for cystic fibrosis caused by nonsense mutations and therefore is of broad interest for the audience of Nature Communications. Please find our point-by-point responses to the specific comments below.

Impact of the main conclusion: The scientific value of the main finding — all the tested CFTR nonsense mutations are SMG6-dependent — is not clear. How often NMD is mediated by the SMG6 pathway, as opposed to the SMG5/7 pathway? Is it 10% or 90%? If it is the former, the finding would have had greater scientific implications because readers, if it wasn’t for this study, would have assumed that they are dependent on the SMG5/7 pathway. Such conclusion would also have greater clinical implications too because, if the SMG6 pathway controls only a small fraction of genes, therapeutically knocking down the pathway would have smaller side effects. However, recent studies have suggested that the SMG6-

mediated endonucleolytic cleavage pathway is likely a more prevalent mechanism of NMD than the SMG5/7-mediated pathway, as the loss of SMG6 impaired NMD more severely than the loss of SMG7 (Boehm et al., 2021 Nat Comm). If the SMG6 pathway is the main pathway anyway, the fact that CFTR nonsense mutations are dependent on SMG6 have only limited impact.

Response: We thank the reviewer for carefully reviewing our work. However, we respectfully disagree with this conclusion regarding the impact of the discovery of SMG6-dependent regulation of CFTR mRNAs. The reviewer referenced (Boehm et al., 2021 Nat Comm) to support their statement “recent studies have suggested that the SMG6-mediated endonucleolytic cleavage pathway is likely a more prevalent mechanism of NMD than the SMG5/7-mediated pathway, as the loss of SMG6 impaired NMD more severely than the loss of SMG7.” However, the key conclusion from this publication entitled “SMG5-SMG7 authorize nonsense-mediated mRNA decay by enabling SMG6 endonucleolytic activity” is that the two decay pathways are interlinked as the loss of SMG5-SMG7-dependent pathway also inactivates the SMG6-dependent pathway. This publication does not provide evidence related to the prevalence of either the SMG6-mediated or SMG5/7-mediated pathway for the decay of NMD substrates.

Furthermore, it is still largely unknown how the SMG6 and SMG5/7 decay pathways functionally cooperate to degrade mRNAs and how often NMD is arbitrated by either the SMG6 pathway or the SMG5/7 pathway, although there are several studies which have touched upon this topic. For example, Dr. Oliver Muhlemann’s lab demonstrated using reporter assay that PTC-containing immunoglobulin μ transcripts were preferentially subjected to SMG6-mediated endonucleolytic cleavage whereas β -Globin transcripts were predominantly degraded by the SMG5/SMG7-dependent pathway (Metze et al., 2013 RNA). Then later, this group performed transcriptome analysis of HeLa cells with SMG6 and SMG7 knockdown to identify endogenous NMD substrates regulated by these two NMD branches and demonstrated that SMG6 and SMG7 act on essentially the same transcripts, indicating extensive redundancy between the endo- and exonucleolytic decay routes (Colombo et al., 2017 RNA). An early study from Dr. Torben Jensen’s lab identified wide-spread SMG6-mediated endonucleolysis on endogenous NMD substrates using CAGE-seq (Lykke-Andersen et al., 2014 Genes Dev). However, this report lacked the comparison of SMG5/7-mediated decapping events due to technical difficulties of direct measurement of decapping products. In summary, there is no evidence indicating which decay pathway is more prevalent for NMD, nor information of the percentage of transcripts that are being degraded by the SMG6- or SMG5/7-dependent NMD branch or both. For specific mRNAs with PTCs, the only way to know how the transcript is being degraded is by experimental testing as performed in this study for several cystic fibrosis-causing CFTR nonsense mutations.

Therefore, our finding that all tested CFTR mRNAs with PTCs are degraded by SMG6-mediated endonucleolytic pathway, but not SMG5/7-mediated exonucleolytic pathway is significant and is of high impact. Furthermore, our study is the first to investigate the PTC positional effects on the usage of the NMD branches and therefore merit publication in a high impact journal as Nature Communications.

Therapeutic potential: The authors mentioned that the SMG6-targeting gapmer ASOs could be a therapeutic strategy for CF. While it is agreeable that targeting a branch-specific NMD factor, such as SMG6, would have less side effects than targeting universal factors, such as UPF1 or SMG1. However, targeting SMG6 would still lead to changes in expression of a sizable number of genes (it has not been mentioned in the manuscript how many normal genes are differentially regulated by SMG6 knock-

down). Therefore, therapeutic use of SMG6 gapmer ASOs can lead to significant side effects, and unless the authors experimentally demonstrate the safety of the gapmer ASOs (which is not given in the manuscript), it is hard to justify the claim that the gapmer ASOs are suitable drug candidates. This is a key issue in terms of the impact of this study, because if the gapmer ASOs were to have questionable therapeutic implications, a relatively simpler study using siRNAs for the NMD factors would have led to the same conclusions as the current study. (siRNA experiments are simpler because pre-designed and knock-down guaranteed siRNAs for each gene are readily commercially available off the shelf.) Another point to consider when assessing the clinical impact of this study is that ASO approaches targeting exon junction complex formation of a specific target gene to inhibit NMD of a specific target gene have been published. This alternative approach seems conceptually superior to the approach described in this study because of the specificity, hence potentially lower side effects. Another issue that makes it difficult to assess the clinical significance of this work is that in Fig. 7, the authors showed that the ASOs, when used in combination with a readthrough drug, can restore the CFTR function to some extent. However, since no positive control (i.e., CFTR carrier cells) was performed, it is hard to assess whether or not the improved CFTR function is going to provide meaningful clinical improvement.

Response: We believe our finding demonstrating that all tested CFTR nonsense mutations are regulated by the branch specific NMD factor SMG6 significantly advances the knowledge of how the NMD pathway degrades CFTR mRNAs with nonsense codons. This finding provides direction for the development of therapeutic NMD inhibitors for CF caused by nonsense mutations. The therapeutic strategy of NMD pathway inhibition has been explored for decades. As mentioned by this reviewer, the discovery of NMD branches opened a possibility for targeting branch-specific NMD factors for disease caused by nonsense mutations, since targeting these factors would have less negative effects than targeting the core factors such as UPF1 or SMG1. However, the branch responsible for the degradation of CFTR mRNAs harboring nonsense codons is currently unknown. Our discovery has filled this gap in knowledge. We agree with this reviewer that targeting a branch specific NMD factor, such as SMG6, will still likely result in gene expression changes. However, it is not known if SMG6 inhibition will have deleterious effects in human lung, and whether these effects can be tolerated for a devastating disease such as cystic fibrosis. As such, future work is warranted to further evaluate the therapeutic potential of SMG6 inhibition for CF caused by nonsense mutations, which is beyond the scope of this study. We have softened the phrasing related to therapeutic implications and added additional discussion in the revised manuscript so that we would not overstate the implications of our findings (line 319-334).

Additionally, although we agree with this reviewer that gapmer ASOs targeting NMD factors require extensive evaluation for clinical applications, we disagree that gapmer ASOs as a drug platform have questionable therapeutic implications. Gapmer ASOs are a clinically validated drug discovery platform which are safe and efficacious modalities for RNA-targeted therapeutics (Crooke et al.; 2021 JCB; Bennett et al.; 2017 ANNU REV PHARMACOL; Bennett et al.; 2019 ANNU REV PHARMACOL). Even though siRNAs are commercially available, use of gapmer ASOs were critical for this study due to several unique attributes of ASOs. For example, the free uptake of unformulated ASOs in vitro facilitated treatment of 16HBE and primary hBE cells cultured at both liquid-liquid or air-liquid interface on Transwell inserts for CFTR function assessment by the Ussing chamber assay.

Finally, we agree with this reviewer that the degree of CFTR upregulation relative to WT CFTR function is important for clinical relevance. We have included additional analysis in figure 7 showing relative % of WT CFTR function from studies in CFF-16HBEge cells. The focus of this study was to evaluate NMD pathway inhibition for the upregulation of CFTR mRNAs with nonsense codons. The degree of CFTR functional upregulation in 16HBEge cells relative to WT-CFTR is likely limited by the efficacy of the translational readthrough agent G418, not the strategy of NMD pathway inhibition. We have added additional discussion points related to the challenges and recent advances in the development of translational readthrough agents which can be combined with NMD pathway inhibitors (line 307-318).

Mutation selection: When the authors could have chosen any nonsense mutations of CFTR (mutation-harboring cell lines can be generated by CRISPR knock-in as done in this study, or for common mutations, patient-derived cell lines can be obtained from the foundation), they chose CFTR Y122X, G542X, R1162X, or W1282X. An issue here is that they did not explain the selection criteria. While G542X and W1282X are reasonable choices in that they are high frequency mutations (2.5% and 1.2% allele frequency in CF patients (https://cftr2.org/mutations_history)) Y122X has a very low frequency (0.06%). It is also not explained why R553X is not chosen in this study, when it has a 10 times higher frequency (0.9%) than Y122X. This study highlights Y122X mutation because it is degraded by a different NMD pathway compared to other CFTR nonsense mutations, but it has not been mentioned that the clinical importance of this mutation is limited as it has a low allele frequency in patients. In addition, the authors did not provide any mechanistic basis for the UPF2/3-independent decay of Y122X transcripts (i.e., what are the determinants of pathway selection).

Response: We thank the reviewer for this comment. We selected the mutations based on the position in the CFTR mRNA. To date, there are 67 different CFTR nonsense mutations reported (CFTR2 database). Evidence suggests that the position of the nonsense mutation in a gene could influence the gene expression outcome at the level of mRNA and protein due to degradation of the mRNAs by NMD and the efficiency of translational readthrough, respectively. Therefore, we hypothesized that the location of the nonsense mutation could influence the usage of different NMD branches. To test this, we grouped CFTR nonsense mutations into three categories based on the resulting position of the PTC in the CFTR protein, as early nonsense mutations – generating PTCs within the first 150 amino acids, middle nonsense mutations – generating PTCs between the 150th and 1000th amino acid, or late nonsense mutations – generating PTCs after 1000th amino acid. We selected the most representative nonsense mutations from each of the categories Y122X (early), G542X (middle), R1162X (late), and W1282X (late) to evaluate NMD regulation of CFTR mRNAs harboring nonsense codons. Y122X was selected as it is the most prevalent early nonsense mutation that is subject to NMD. R553X, as this reviewer mentioned, is a nonsense mutation with a high prevalence (0.9%), but it is located in the same exon with proximity to G542X, which has a much higher prevalence (2.5%). We hypothesize that R553X will be regulated similarly as G542X by NMD and as such was not selected for evaluation. Overall, we believe our selection of nonsense mutations is well justified based on scientific evidence and hypothesis. We thank the reviewer for pointing out the possible confusion and provided additional descriptions for the rationale of mutation-selection in the manuscript (Line 97-108) as well as an additional schematic figure 1A illustrating the selection criteria

REVIEWERS' COMMENTS

Reviewer #1 (Remarks to the Author):

The authors' responses to comments were excellent. The changes made to manuscript and figures therein address all of the previous concerns. This reviewer does not have any further comments.

Reviewer #2 (Remarks to the Author):

The manuscript has been improved by addressing most of the major critiques. The residual concern remains ascertaining clinical significance, a point made as Critique 1 in the original review, regarding the need for primary cells to assess bioactivity and translatability. While primary cells are difficult to obtain, there are repositories available, including a bank of nasal cells from CF patients provided freely to investigators for use by the CF Foundation. These experiments should be performed with the key SMG inhibition conditions to allow results to be contextualized. The assessment of organoids, available via other repositories, could be an alternative assessment strategy.

Reviewer #3 (Remarks to the Author):

The authors have addressed/explained most of the critical issues pointed out by the reviewers.